# Temporal Variational Implicit Neural Representations

**Batuhan Koyuncu**                                                    *koyuncu@cs.uni-saarland.de*
*Department of Computer Science, Saarland University*

**Rachael DeVries**                                                    *rachael.devries@bio.ku.dk*
*Department of Biology, University of Copenhagen*
*Novo Nordisk A/S*

**Ole Winther**                                                        *ole.winther@bio.ku.dk*
*Department of Biology, University of Copenhagen*
*Department of Applied Mathematics and Computer Science, Technical University of Denmark*

**Isabel Valera**                                                      *ivalera@cs.uni-saarland.de*
*Department of Computer Science, Saarland University*

**Reviewed on OpenReview:** *https://openreview.net/forum?id=1CGfvw4ySe*

## Abstract

We introduce Temporal Variational Implicit Neural Representations (TV-INRs), a probabilistic framework for modeling irregular multivariate time series that enables efficient and accurate individualized imputation and forecasting. By integrating implicit neural representations with latent variable models, TV-INRs learn distributions over time-continuous generator functions conditioned on signal-specific covariates. Unlike existing INR approaches that require extensive training, fine-tuning or meta-learning, our method achieves accurate individualized predictions through a single forward pass. Our experiments demonstrate that with a single TV-INRs instance, we can accurately solve diverse imputation and forecasting tasks, offering a computationally efficient and scalable solution for real-world applications. TV-INRs performs particularly well in low-data regimes, where on several datasets it achieves substantially lower imputation error, including order-of-magnitude improvements.

## 1 Introduction

Time series are a key way to represent data in many domains, from energy consumption to finance, and they frequently contain missing values and irregularities due to sensor malfunctions, collection errors, or resource constraints (Che et al., 2018; Du et al., 2023; Proietti & Pedregal, 2023). These challenges are particularly pronounced in clinical datasets, which often exhibit extreme sparsity (80-90% missingness) and noisy, irregular sampling due to human involvement in non-automated measurements (Silva et al., 2012). In order to impute missing values and forecast future time points, effective solutions must handle these challenges while utilizing available covariates to capture unique temporal dynamics.

Current methods relying on Recurrent Neural Networks (RNNs) (Chung et al., 2015; Che et al., 2018) and Transformers (Bansal et al., 2023; Liu et al., 2023) are generally tailored for regular, dense time series data and require placeholders for missing observations. They also operate in discrete time, and careful design is necessary for continuous time settings (Chen et al., 2024). Alternatively, there exist continuous time series models which use Implicit Neural Representations (INRs) (Sitzmann et al., 2020) to handle irregular time series data (Naour et al., 2024; Cho et al., 2024). By learning a unique continuous function to represent each time series, INRs have great potential for individualization by capturing the unique activity patterns of each subject. However, existing approaches are inflexible, and often require training multiple models, fine-tuning, or meta-learning to handle variations in data availability, prediction length, and individualization. For example, the method presented in Naour et al. (2024) requires the training of separate models for different missingness ratios or horizon lengths, and performs gradient-based meta-learning during inference, resulting

in a data-hungry model. Such approaches are impractical in real-world applications where scalability and generalization are crucial, as computational resources may be limited during deployment.

To address these shortcomings, we introduce Temporal Variational Implicit Neural Representations (TV-INRs), a novel probabilistic model for multivariate time series with INRs. We use INRs as generator functions for continuous time series modeling, effectively handling the challenge of irregular sampling. By also integrating latent variable models and amortized variational inference, TV-INRs learns distributions over INRs conditioned on individual signals and their covariates through a learned latent space. This approach is therefore scenario and sample agnostic, accommodating varying levels of missingness or time series length and eliminating the need for task-specific retraining or per-sample optimization. In short, we preserve the benefits of INRs for time series while making them scalable and efficient. Our model pushes forward multivariate time series analysis with several key contributions:

- We introduce a fully probabilistic framework for multivariate time series based on implicit neural representations.
- TV-INRs achieves competitive accuracy to gradient-based meta-learning approaches and improves imputation performance in low-data scenarios, while avoiding per-sample optimization during inference.
- We demonstrate successful generalization across multiple data settings, including missingness and forecasting horizon length, with a single training. This significantly reduces training requirements relative to comparable models.
- Our results show that the inclusion of covariates enables effective individualization and further increases our model's accuracy with sparse data, demonstrating suitability for real-world applications with extreme missingness, such as healthcare.

## 2 Related work

### 2.1 Learning implicit neural representations

**Hypernetworks** denoted as $g_\phi$, are neural networks that generate parameters $\theta = g_\phi(\cdot)$ for another neural network $f_\theta(\cdot)$ (Ha et al., 2016). Hypernetworks can generate task-specific model parameters, making them suitable for meta-learning scenarios that require quick adaptation to new tasks. Zhao et al. (2020) showed that meta-learning a hypernetwork effectively modulates inner-loop optimization and adapts features task-dependently using model-agnostic meta-learning. Nguyen et al. (2022) proposed to generate parameters of the approximate posterior and likelihood of a Variational Autoencoder (VAE) model to perform multiple tasks. Recent works have shown hypernetworks to be useful for generating parameters for implicit neural representations (Dupont et al., 2021; Koyuncu et al., 2023).

**Implicit neural representations (INRs)** offer a novel approach to data representation and modeling complex continuous signals using the weight space (Sitzmann et al., 2020). This formulation is supported by strong theoretical guarantees and makes the model inherently resolution-agnostic and robust to irregular sampling (Sitzmann et al., 2020). By leveraging neural networks, particularly multi-layer perceptrons (MLPs), represented as $f_\theta(\cdot)$, INRs effectively map coordinates to features like color, occupancy, or amplitude. Therefore INRs enable continuous representation of high-dimensional data, offering significant advantages in various domains, including images, 3D shape modeling, spatio-temporal data (Dupont et al., 2021; 2022a; Koyuncu et al., 2023; Park et al., 2024) and geometric structures (Vetsch et al., 2022; Niemeyer et al., 2022), because predictions are not constrained by input range or resolution. Recent works are actively exploring parameterization strategies for INRs. For example, approaches by Dupont et al. (2022b); Strümpler et al. (2022) have used compressed representations of the data as inputs to hypernetworks $g_\phi$, which then generate weights $\theta$ of the INRs $f_\theta(\cdot)$. Peis et al. (2025) uses latent diffusion models to generate a latent variable model to model the weights of INRs via a transformer network. And Park et al. (2024) proposed to learn sample-specific dynamic positional embeddings, rather than modeling INRs weights.

**Meta-learning** is a learning approach where algorithms are designed to improve their learning efficiency and adaptability across different tasks and domain shifts. In model-agnostic meta-learning (MAML), the aim is to fine-tune the trained model using test instances with gradient updates (Finn & Levine, 2017; Wang et al., 2020). This is particularly relevant in scenarios when adaptation of the model is needed for unseen

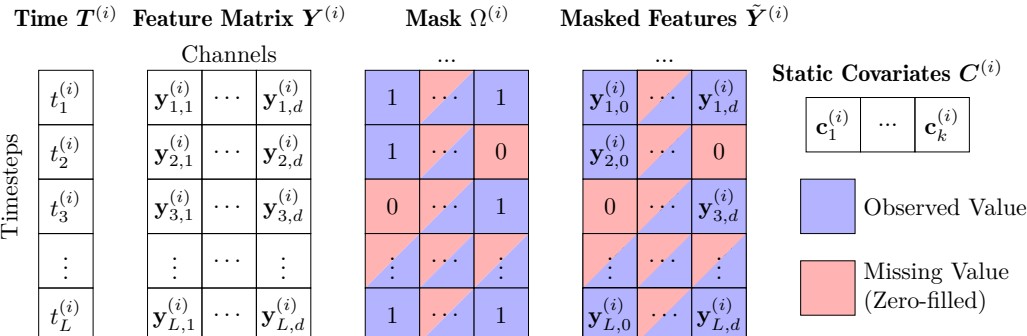

Figure 1: Visualization of temporal stamps $\boldsymbol{T}$, features $\boldsymbol{Y}$, mask $\Omega$, and static covariates $\boldsymbol{C}$. $\boldsymbol{T}$ and $\boldsymbol{Y}$ represent the input signal, $\Omega$ indicates missing values with binary entries, and $\boldsymbol{C}$ contains time-invariant covariates.

data during inference. MAML is widely used to update INR weights (Dupont et al., 2022a; Jeong & Shin, 2022; Niemeyer et al., 2022; Bamford et al., 2023), however, its reliance on a test-time optimization step for each sample introduces computational overhead scaling with the number of test instances.

## 2.2 Time series imputation and forecasting

RNNs are frequently used for time series forecasting, due of their ability to capture sequential dependencies (Chung et al., 2015; Hewamalage et al., 2021; Che et al., 2018; Guo et al., 2016). However, they assume fixed frequencies and struggle with long-term dependencies. To address these limitations, LSTM networks incorporate memory cells that retain relevant historical information while discarding irrelevant data (Hochreiter, 1997; Hua et al., 2019; Chen et al., 2022). Recent advancements have also embraced transformer-based architectures for time series modeling. Models such as SAITS (Du et al., 2023), PatchTST (Nie et al., 2023) and iTransformer (Liu et al., 2023) leverage attention and embedding strategies to capture both short- and long-term time dependencies within time series. Despite their strengths, transformers are inherently discrete and may fail to interpolate between time steps unless they are carefully redesigned for this task (Chen et al., 2024). Moreover, they may have trouble identifying and preserving key information when attending to large inputs (Wen et al., 2022). Likewise, conditional diffusion models like CSDI (Tashiro et al., 2021) and LSCD (Fons et al., 2025) operate on fixed temporal grids and rely on architectural workarounds to manage irregular observations.

Recently, INRs have been used in continuous modeling of time series data for imputation and forecasting tasks (Naour et al., 2024; Fons et al., 2022; Cho et al., 2024), and for anomaly detection (Jeong & Shin, 2022). Fons et al. (2022) use a set-encoder approach to generate latent representations to parameterize INRs through hypernetworks for time series generation. Similarly, Bamford et al. (2023) adopt this approach for time series imputation, utilizing an auto-decoding strategy that requires back-propagation to learn these latent representations. Naour et al. (2024); Cho et al. (2024); Woo et al. (2023) use gradient-based meta-learning approaches to learn per instance modulations on INRs to perform imputation and forecasting on test data. Therefore, these methods encounter scalability challenges with an increasing number of test instances, since each requires per-instance optimization, and they may underperform in scenarios characterized by limited data availability. Finally, recent work (Naour et al., 2025) explores large-scale INR-based temporal foundation models designed to support out-of-distribution generalization. While this direction is highly complementary to our approach, evaluating foundation-scale pretraining or explicit out-of-distribution generalization lies beyond the scope of our experimental setting, which focuses on in-distribution generalization across varying missingness levels and forecasting horizons.

## 3 Temporal variational implicit neural representations

In this section, we introduce **T**emporal **V**ariational **I**mplicit **N**eural **R**epresentations (TV-INRs). Our approach is motivated by representing time series as continuous functions using Implicit Neural Representations (INRs). Leveraging the amortized inference framework of Variational Autoencoders (Kingma, 2013; Rezende et al., 2014), TV-INRs learns distributions over INR parameters through encoder networks, eliminating per-sample optimization during inference while enabling efficient scaling to large datasets (Cremer et al., 2018; Hoffman

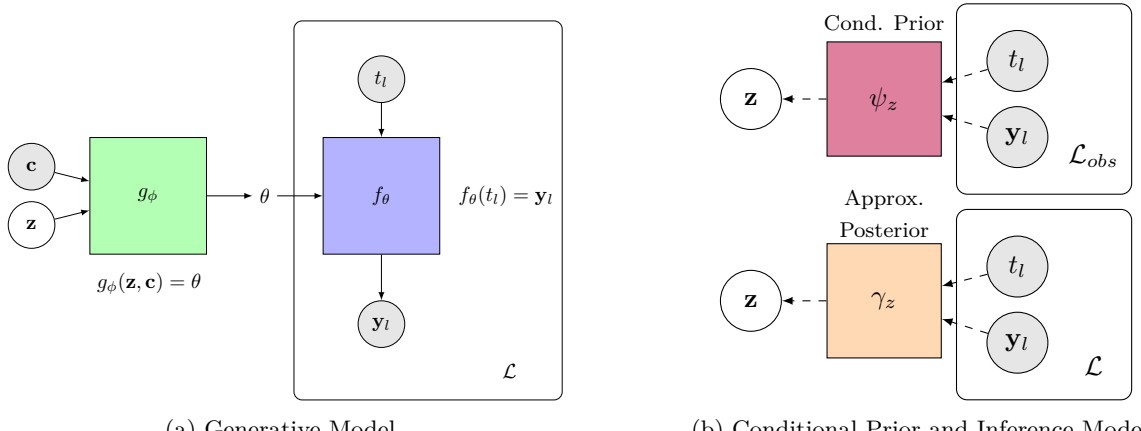

(a) Generative Model          (b) Conditional Prior and Inference Model

Figure 2: Graphical models for generative and inference tasks.

et al., 2013; Mnih & Gregor, 2014). This approach maintains competitive performance for time series modeling tasks such as imputation and forecasting while facilitating personalized modeling through latent variables.

**Notation.** Let $[L] = \{1, \ldots, L\}$ denote the set of positive integers from 1 to $L$ and $d$ denote the total number of feature dimensions. We consider a dataset of $N$ samples $\{(\boldsymbol{T}^{(i)}, \boldsymbol{Y}^{(i)}, \boldsymbol{C}^{(i)})\}_{i=1}^N$, where each sample $i \in [N]$ as shown in Fig. 1 includes:

- **Temporal stamps**: A point cloud of $L_i$ temporal stamps (i.e. *temporal* coordinates), $\boldsymbol{T}^{(i)} = \{t_l^{(i)}\}_{l=1}^{L_i}$, with $t \in \mathbb{R}$.
- **Feature vectors**: Corresponding feature vectors $\boldsymbol{Y}^{(i)} = \{\mathbf{y}_l^{(i)}\}_{l=1}^{L_i}$, where $\mathbf{y}_l^{(i)} \in \mathbb{R}^{d_l^{(i)}}$ with $d_l^{(i)} \leq d$ representing the number of observed channels at index $l$. The set $\mathcal{A}^{(i)}$ identifies indexes ($l$) where channels ($j$) are absent in the original dataset.
- **Static covariates**: Static covariates $\boldsymbol{C}^{(i)} = \{\mathbf{c}^{(i)}\}$, where $\mathbf{c} \in \mathbb{R}^k$, which are constant for all stamps in the sample.

We denote the multichannel $i$-th time series as a tuple $\boldsymbol{X}^{(i)} = (\boldsymbol{T}^{(i)}, \boldsymbol{Y}^{(i)})$, consisting of $L_i$ (irregular) temporal stamps and their corresponding features. To effectively handle missing data, we distinguish between three sets of indices. The observed indices $\mathcal{O}^{(i)}$ represent available data points in our dataset, which we input to the model. The masked indices $\mathcal{M}^{(i)}$ correspond to entries we artificially mask during training to facilitate self-supervised learning and improve generalization to missing data scenarios (Moreno-Muñoz et al., 2023). Finally, the absent indices $\mathcal{A}^{(i)}$ are inherent to the data and represent entries of missing channels due to partial observations or limitations in data collection which we exclude from the training process as they represent inherent data incompleteness rather than synthetic masks. We define a binary mask $\Omega^{(i)}$ to formalize this as:

$$\Omega_{l,k}^{(i)} = \begin{cases} 1 & \text{if } (l,k) \in \mathcal{O}^{(i)} \\ 0 & \text{if } (l,k) \in \mathcal{M}^{(i)} \\ 0 & \text{if } (l,k) \in \mathcal{A}^{(i)} \end{cases} \tag{1}$$

where $\mathcal{O}^{(i)}, \mathcal{M}^{(i)}, \mathcal{A}^{(i)} \subseteq [L_i] \times [d]$ with $\mathcal{O}^{(i)} \cap \mathcal{M}^{(i)} = \emptyset$. Finally, we denote by $\tau$ the percentage of observed indices in the available data, i.e., $\tau = \frac{|\mathcal{O}^{(i)}|}{|\mathcal{O}^{(i)} \cup \mathcal{M}^{(i)}|}$.

### 3.1 Model description

**Generative model.** To ease readability, we consider the model for a single sample and omit the use of the superscript ($i$). TV-INRs is generative model for the feature set $\boldsymbol{Y}$ given timestamps $\boldsymbol{T}$. For now, we assume that $(\boldsymbol{T}, \boldsymbol{Y})$ is a timeseries with $L$ elements and $d$ channels without any absence, e.g. $\mathcal{A} = \emptyset$. The observed data $\boldsymbol{Y}_{\text{obs}}$ indexed by $\mathcal{O}^{(i)}$ and corresponding timestamps $\boldsymbol{T}_{\text{obs}}$ are given as context to the model, while $\boldsymbol{Y}_{\text{m}}$ indexed by $\mathcal{M}^{(i)}$ represents the masked values to predict at given timestamps $\boldsymbol{T}_{\text{m}}$. Together, they form the

complete datasets: $\boldsymbol{Y} = \boldsymbol{Y}_{\mathrm{m}} \cup \boldsymbol{Y}_{\mathrm{obs}}$ and $\boldsymbol{T} = \boldsymbol{T}_{\mathrm{m}} \cup \boldsymbol{T}_{\mathrm{obs}}$ with the assumption of $\mathcal{A} = \emptyset$. The joint distribution can be written in a general form

$$p(\boldsymbol{Y}_{\mathrm{m}}, \boldsymbol{Y}_{\mathrm{obs}}, \mathbf{z}|\boldsymbol{Y}_{\mathrm{obs}}, \boldsymbol{T}, \mathbf{c}) = p_{\boldsymbol{\psi}_{\mathbf{z}}}(\mathbf{z}|\boldsymbol{Y}_{\mathrm{obs}}, \boldsymbol{T}_{\mathrm{obs}}) \prod_{l=1}^{L} p_{\boldsymbol{\theta}(\mathbf{z},\mathbf{c})}(\mathbf{y}_l|t_l) \tag{2}$$

where $\mathbf{z}$ represents a latent variable and $\mathbf{c}$ denotes covariates. To generate such a signal, the process begins by sampling a continuous latent variable $\mathbf{z}$ from a conditional prior distribution, $p_{\boldsymbol{\psi}_{\mathbf{z}}}(\mathbf{z}|\boldsymbol{Y}_{\mathrm{obs}}, \boldsymbol{T}_{\mathrm{obs}}) = \mathcal{N}(\mathbf{z}|f_{\boldsymbol{\psi}_{\mathbf{z}}}(\boldsymbol{Y}_{\mathrm{obs}}, \boldsymbol{T}_{\mathrm{obs}}))$, which is parameterized by $\boldsymbol{\psi}_{\mathbf{z}}$ using a Transformer encoder. The resulting vector $\mathbf{z}$, concatenated with random variable $\mathbf{c}$, acts as input to the *hypergenerator*. Here, the hypergenerator is an MLP-based hypernetwork $g_{\phi_k}(\mathbf{z}, \mathbf{c})$, with input $[\mathbf{z}, \mathbf{c}]$ that outputs a set of parameters $\boldsymbol{\theta}_k = g_{\phi_k}(\mathbf{z}, \mathbf{c})$; and, a data generator, $f_{\theta}$, parametrized by the output of the hypernetwork. Thus, both $\mathbf{z}$ and $\boldsymbol{\theta}$ encode the information shared among the stamps in the data (e.g., features) generation process as shown in Fig. 2a. Moreover, we refer to TV-INRs as C-TV-INRs when covariates are available and used.

**Inference model.** We approximate posterior distribution as $q_{\boldsymbol{\gamma}_z}(\mathbf{z}|\boldsymbol{Y}, \boldsymbol{T}) = \mathcal{N}(\mathbf{z}|f_{\boldsymbol{\gamma}_z}(\boldsymbol{Y}, \boldsymbol{T}))$, parameterized by $\boldsymbol{\gamma}_z$. It's important to note that this distribution is shared among the complete instance (e.g., time series signal), thus $\mathbf{z}$ contains global information as shown in Fig. 2b.

**Training.** We employ masked training by maximizing the evidence lower bound (ELBO) of the proposed model, which is given by

$$\mathcal{L}(\boldsymbol{T}, \boldsymbol{Y}, \boldsymbol{C}) = \mathbb{E}_{q_{\gamma}} \left[ \log p_{\boldsymbol{\theta}(\mathbf{z},\mathbf{c})}[\boldsymbol{Y} \mid \boldsymbol{T}] \right] - D_{\mathrm{KL}} \left( q_{\boldsymbol{\gamma}_z}(\mathbf{z} \mid \boldsymbol{Y}, \boldsymbol{T}) \| p_{\boldsymbol{\psi}_{\mathbf{z}}}(\mathbf{z}|\boldsymbol{Y}_{\mathrm{obs}}, \boldsymbol{T}_{\mathrm{obs}}) \right) \tag{3}$$

where $p_{\boldsymbol{\psi}_{\mathbf{z}}}$ and $q_{\boldsymbol{\gamma}_z}$ are Gaussian distributions, and we model $p_{\boldsymbol{\theta}(\mathbf{z},\mathbf{c})}$ with a Laplace distribution as it demonstrates better performance in capturing high-frequency components.

**TV-INRs pipeline summary.** For clarity, we summarize the end-to-end modeling and inference procedure:

- **Input encoding:** Observed values, timestamps, and binary masks are embedded using spatial encoding and Fourier-feature temporal encodings (Fig 1).
- **Latent inference:** A Transformer encoder produces a global latent representation $z$ via amortized variational inference, conditioned on observed data (and covariates when available) (Fig 2b).
- **Hypernetwork generation:** The latent code $z$, concatenated with covariates $c$, is passed through a hypernetwork to generate INR parameters $\theta$ (Fig 2a).
- **Continuous-time decoding:** The INR $f_{\theta}(t)$ produces distributional predictions for any queried timestamp, enabling imputation and forecasting in continuous time (Fig 2a).

In the next section, we provide implementation details for each step.

### 3.2 Implementation details

We model the conditional prior and approximate posterior with Transformer encoders. To handle heterogeneity in the input data, we augment the input features by concatenating them with a binary mask, ($\Omega^{(i)} \in 0, 1^{L_i \times d}$), which indicates observed entries across both temporal and feature dimensions.

**Input processing.** For each sample $i \in [N]$, we process the input tuple $(\boldsymbol{T}^{(i)}, \boldsymbol{Y}^{(i)}, \boldsymbol{C}^{(i)})$ to handle missing values. We construct the input representation using the binary mask ($\Omega^{(i)}$) as follows:

1. Fill masked values in $\boldsymbol{Y}^{(i)}$ with zeros:

$$\tilde{\boldsymbol{Y}}_{l,k}^{(i)} = \begin{cases} \boldsymbol{Y}_{l,k}^{(i)} & \text{if } (l, k) \in \mathcal{O}^{(i)} \\ 0 & \text{if } (l, k) \in \mathcal{U}^{(i)} \end{cases} \tag{4}$$

where $\tilde{\boldsymbol{Y}}^{(i)} \in \mathbb{R}^{L_i \times d}$, in case for the input of the posterior encoder we give full available data.

2. Concatenate the mask along the feature dimension and transform the processed features with a linear layer for spatial encoding, which captures relationships among different channels, yielding $\boldsymbol{E}_{\mathrm{spatial}}^{(i)} = f_{\mathrm{linear}}(\bar{\boldsymbol{Y}}^{(i)}) \in \mathbb{R}^{L_i \times d_{\mathrm{model}}}$, where $\bar{\boldsymbol{Y}}^{(i)} = [\tilde{\boldsymbol{Y}}^{(i)}; \Omega^{(i)}] \in \mathbb{R}^{L_i \times 2d}$.

3. Expand temporal coordinates with channel indices $\mathbf{v}_d = [0, ..., d-1]$ and encode them with Fourier Features (FoF) (Dupont et al., 2021): $\boldsymbol{E}_{\mathrm{temporal}}^{(i)} = \mathrm{FoF}(\bar{\boldsymbol{T}}^{(i)}) \in \mathbb{R}^{L_i \times d_{\mathrm{model}}}$, where $\bar{\boldsymbol{T}}^{(i)} = \boldsymbol{T}^{(i)} \otimes \mathbf{v}_d \in \mathbb{R}^{L_i \times d}$.

The final embedding $\boldsymbol{E}^{(i)} = \boldsymbol{E}^{(i)}_{\text{spatial}} + \boldsymbol{E}^{(i)}_{\text{temporal}}$ is element-wise summed and then fed into the encoder.

**Encoding.** The embedded input $\boldsymbol{E}^{(i)}$ is processed through a transformer encoder to model the conditional distributions $p_{\boldsymbol{\psi_z}}(\mathbf{z}|\boldsymbol{Y}_{\text{obs}}, \boldsymbol{T}_{\text{obs}})$ and $q_{\boldsymbol{\gamma_z}}(\mathbf{z}|\boldsymbol{Y}, \boldsymbol{T})$. The encoder takes $\boldsymbol{E}^{(i)}$, transforms the input through self-attention, applies pooling (POOL) over temporal dimension, and a feed-forward network (FFN) generates parameters to model the latent features $\mathbf{z}$:

$$\mathbf{z} \sim \mathcal{N}(\boldsymbol{\mu}, \boldsymbol{\Sigma}) \text{ where } \boldsymbol{\mu}, \boldsymbol{\sigma} = \text{FFN}(\text{POOL}(\boldsymbol{H})), \text{ and } \boldsymbol{H} = \text{Transformer}(\boldsymbol{E}^{(i)}) \tag{5}$$

where $\boldsymbol{\Sigma} = \text{diag}(\boldsymbol{\sigma}^2)$. Here, we make sure masked values are not used during attention computation.

**Decoding.** The latent representation ($\mathbf{z}$) is combined with conditional variables to construct the decoder input through the following steps:

1. The conditional variables $\boldsymbol{C}^{(i)}$ are first binned and then transformed by a feed-forward network into $\bar{\mathbf{c}} = \text{FFN}(\boldsymbol{C}^{(i)}) \in \mathbb{R}^{d_c}$, which is subsequently concatenated with the latent representation to form the decoder input $\boldsymbol{h}_{\text{dec}} = [\mathbf{z}; \bar{\mathbf{c}}]$.

2. The resulting $\boldsymbol{h}_{\text{dec}}$ is passed through a hypernetwork $g_\phi$ to generate the parameters $\theta = g_\phi(\boldsymbol{h}_{\text{dec}})$ for the implicit neural representation (INR), $f_\theta$, which is continuous over $t$ (Sitzmann et al., 2020).

3. The INR, $f_\theta$, models the output feature values as $\hat{\boldsymbol{y}}_l \sim \text{Laplace}(\mu_l, b_l)$, where the distribution's parameters $(\mu_l, b_l) = f_\theta(\boldsymbol{e}_l)$ are the output of mapping the encoded time point $\boldsymbol{e}_l$.

## 4 Experiments

**Baselines.** We thoroughly tested TV-INRs framework across imputation and forecasting tasks in full and limited data regimes with uni- and multi-variate datasets. We compare our model with TimeFlow (Naour et al., 2024), an INR-based time series model. It requires training separate models for different missingness ratios or horizon lengths, and performs gradient-based meta-learning during inference (details in App. A.8). We include two baselines specifically designed for time series imputation: SAITS (Du et al., 2023), which is based on self-attention, and CSDI (Tashiro et al., 2021), a conditional diffusion model that operates on a fixed temporal grid. For the forecasting task, we compare with DeepTime (Woo et al., 2023), which learns deep time-index models specifically designed for time series forecasting. Potential baselines HyperTime (Fons et al., 2022), MADS (Bamford et al., 2023), LSCD (Fons et al., 2025) were not available as full open-source models, and were therefore not tested.

**Univariate datasets.** We conducted experiments on four univariate datasets (App. A.2 Table 5), and compared our approach to Timeflow (Naour et al., 2024), DeepTime (Woo et al., 2023), SAITS (Du et al., 2023), and CSDI (Tashiro et al., 2021). Each dataset comprises one-dimensional signals originating from various locations or sources, and is available at the Monash Time Series Forecasting repository (Godahewa et al., 2021).

**Multivariate datasets.** While some datasets contain regular sampling (e.g., electricity), others are irregular, and have multiple sensors with unique temporal patterns. TV-INRs is the first temporal INR model to handle such multivariate signals, leading us to exclude Timeflow from these comparisons. We conducted experiments on two multivariate datasets, namely, HAR and The PhysioNet Challenge 2012 (P12), and compared our method with SAITS (Du et al., 2023) and CSDI (Tashiro et al., 2021). Additional details on the datasets, including missingness patterns, are provided in App. A.2.

Next, we describe the imputation and forecasting tasks. Let the $i$-th sample, $\boldsymbol{T}^{(i)} = \{t_j^{(i)}\}_{j=1}^{L_i}$, contain $L_i$ stamps. For both tasks, we compare predicted values against the ground truth for test data using Mean Squared Error (MSE) and Mean Absolute Error (MAE).

**Imputation task.** We partition the data based on an observed ratio $\tau$. Given the observed stamps $\boldsymbol{T}^{(i)}_{\text{obs}}$, the goal is to predict features at the unobserved stamps $\boldsymbol{T}^{(i)}_{\text{unobs}}$, where

$$\boldsymbol{T}^{(i)} = \boldsymbol{T}^{(i)}_{\text{obs}} \cup \boldsymbol{T}^{(i)}_{\text{unobs}}, \quad \boldsymbol{Y}^{(i)} = \boldsymbol{Y}^{(i)}_{\text{obs}} \cup \boldsymbol{Y}^{(i)}_{\text{unobs}}, \quad \hat{\boldsymbol{Y}}_{\text{unobs}} \sim p_{\boldsymbol{\theta}(\mathbf{z},\mathbf{c})}(\boldsymbol{Y}_{\text{unobs}} \mid \boldsymbol{T}_{\text{unobs}}). \tag{6}$$

The task's difficulty increases as $\tau$ decreases. For prediction, we use the conditional prior distribution $p_{\boldsymbol{\psi_z}}(\mathbf{z}|\boldsymbol{Y}_{\text{obs}}, \boldsymbol{T}_{\text{obs}})$ and covariates $\mathbf{c}$ (if available).

**Forecasting task.** We partition data at a horizon $t_{\text{horizon}}$ into history and forecast sets. Given the observed historical data $\boldsymbol{Y}_{\text{hist}}^{(i)}$, our task is to predict $\boldsymbol{Y}_{\text{forecast}}^{(i)}$. We use our conditional prior $p_{\boldsymbol{\psi}_{\mathbf{z}}}(\mathbf{z}|\boldsymbol{Y}_{\text{hist}}, \boldsymbol{T}_{\text{hist}})$ and covariates $\mathbf{c}$ (if available) to generate predictions:

$$\boldsymbol{T}_{\text{hist}}^{(i)} = \{t_j^{(i)} \in \boldsymbol{T}^{(i)} \mid t_j^{(i)} \le t_{\text{horizon}}\}, \; \boldsymbol{T}_{\text{forecast}}^{(i)} = \{t_j^{(i)} \in \boldsymbol{T}^{(i)} \mid t_j^{(i)} > t_{\text{horizon}}\} \tag{7}$$

$$\hat{\boldsymbol{Y}}_{\text{forecast}} \sim p_{\boldsymbol{\theta}(\mathbf{z},\mathbf{c})}(\boldsymbol{Y}_{\text{forecast}} \mid \boldsymbol{T}_{\text{forecast}}). \tag{8}$$

## 4.1 Results

In Sections 4.1.1 and 4.1.2, we explore TV-INRs performance in imputation and forecasting on univariate datasets in comparison with the baseline models Timeflow (Naour et al., 2024), SAITS (Du et al., 2023), CSDI (Tashiro et al., 2021) and DeepTime (Woo et al., 2023). We comment on the training efficiency in Sections 4.1.3 and App. A.10. In Section 4.1.4, we report TV-INRs performance on multivariate datasets including the conditional version of our model, C-TV-INRs, compared with SAITS (Du et al., 2023) and CSDI (Tashiro et al., 2021). Complete pairwise statistical significance results for all datasets are reported in Appendix B.5. Ablation studies on the number of Fourier Features and our INR-based decoder are in App. A.12 and A.13, respectively. The code will be accessible in our repository.

### 4.1.1 Imputation on univariate datasets

Table 1: **Univariate imputation results** with signal lengths $L$, training/testing observation rates $\tau_{\text{train,test}}$, and MSE/MAE evaluated on unobserved indices from non-overlapping test signals. Bold indicates the best mean performance and underlined values indicate the second-best mean performance (lower is better).

| Model | L | $\tau_{\text{Train}}$ | $\tau_{\text{Test}}$ | Electricity MSE | Electricity MAE | Traffic MSE | Traffic MAE | L | Solar-10 MSE | Solar-10 MAE |
|---|---|---|---|---|---|---|---|---|---|---|
| SAITS | 2K | 0.80 | 0.50 | $0.569 \pm 0.048$ | $0.542 \pm 0.022$ | $\mathbf{0.251 \pm 0.028}$ | $\mathbf{0.246 \pm 0.015}$ | | $1.086 \pm 0.005$ | $0.648 \pm 0.022$ |
| | | | 0.30 | $0.793 \pm 0.055$ | $0.654 \pm 0.023$ | $\mathbf{0.337 \pm 0.033}$ | $\mathbf{0.306 \pm 0.015}$ | 10K | $1.087 \pm 0.009$ | $0.651 \pm 0.024$ |
| | | | 0.05 | $1.318 \pm 0.051$ | $0.902 \pm 0.025$ | $0.824 \pm 0.040$ | $0.619 \pm 0.014$ | | $1.126 \pm 0.061$ | $\underline{0.676 \pm 0.062}$ |
| CSDI | 2K | $\sim \mathcal{U}$ | 0.50 | $2.070 \pm 0.194$ | $1.033 \pm 0.023$ | $1.150 \pm 0.029$ | $0.773 \pm 0.144$ | | $1.275 \pm 0.382$ | $0.699 \pm 0.781$ |
| | | | 0.30 | $2.287 \pm 0.157$ | $1.045 \pm 0.012$ | $1.146 \pm 0.103$ | $0.773 \pm 0.165$ | 10K | $1.285 \pm 0.191$ | $0.703 \pm 0.749$ |
| | | | 0.05 | $1.742 \pm 0.265$ | $1.050 \pm 0.013$ | $1.139 \pm 0.111$ | $0.773 \pm 0.171$ | | $1.279 \pm 0.020$ | $0.700 \pm 0.737$ |
| TimeFlow | 2K | 0.50 | 0.50 | $\mathbf{0.131 \pm 0.011}$ | $\mathbf{0.252 \pm 0.010}$ | $\underline{0.346 \pm 0.036}$ | $\underline{0.369 \pm 0.017}$ | | $\mathbf{0.710 \pm 0.040}$ | $\mathbf{0.617 \pm 0.056}$ |
| | | 0.30 | 0.30 | $\mathbf{0.166 \pm 0.012}$ | $\mathbf{0.288 \pm 0.011}$ | $\underline{0.390 \pm 0.042}$ | $\underline{0.388 \pm 0.018}$ | 10K | $\mathbf{0.812 \pm 0.128}$ | $\mathbf{0.658 \pm 0.121}$ |
| | | 0.05 | 0.05 | $\underline{0.378 \pm 0.034}$ | $\underline{0.458 \pm 0.025}$ | $\underline{0.590 \pm 0.048}$ | $\underline{0.496 \pm 0.020}$ | | $\mathbf{0.833 \pm 0.010}$ | $\mathbf{0.663 \pm 0.096}$ |
| TV-INRs | 2K | $\sim \mathcal{S}$ | 0.50 | $\underline{0.249 \pm 0.019}$ | $\underline{0.331 \pm 0.012}$ | $0.546 \pm 0.022$ | $0.401 \pm 0.015$ | | $\underline{0.955 \pm 0.059}$ | $\underline{0.645 \pm 0.038}$ |
| | | | 0.30 | $\underline{0.250 \pm 0.017}$ | $\underline{0.332 \pm 0.012}$ | $0.551 \pm 0.029$ | $0.403 \pm 0.017$ | 10K | $\underline{0.954 \pm 0.074}$ | $\underline{0.646 \pm 0.050}$ |
| | | | 0.05 | $\mathbf{0.289 \pm 0.019}$ | $\mathbf{0.360 \pm 0.015}$ | $\mathbf{0.570 \pm 0.019}$ | $\mathbf{0.415 \pm 0.013}$ | | $\underline{1.104 \pm 0.265}$ | $0.688 \pm 0.132$ |
| SAITS | 200 | 0.80 | 0.50 | $\underline{0.124 \pm 0.014}$ | $\underline{0.223 \pm 0.010}$ | $\underline{0.230 \pm 0.015}$ | $0.245 \pm 0.008$ | | $\underline{0.066 \pm 0.035}$ | $\underline{0.140 \pm 0.021}$ |
| | | | 0.30 | $\underline{0.231 \pm 0.025}$ | $\underline{0.317 \pm 0.017}$ | $\underline{0.345 \pm 0.019}$ | $\underline{0.320 \pm 0.009}$ | 200 | $\underline{0.099 \pm 0.060}$ | $\underline{0.168 \pm 0.030}$ |
| | | | 0.05 | $\underline{0.937 \pm 0.040}$ | $\underline{0.743 \pm 0.018}$ | $\underline{0.904 \pm 0.020}$ | $\underline{0.641 \pm 0.016}$ | | $\underline{0.564 \pm 0.107}$ | $\underline{0.502 \pm 0.037}$ |
| CSDI | 200 | $\sim \mathcal{U}$ | 0.50 | $1.380 \pm 0.216$ | $0.944 \pm 0.035$ | $1.169 \pm 0.204$ | $0.787 \pm 0.187$ | | $1.010 \pm 0.261$ | $0.602 \pm 0.122$ |
| | | | 0.30 | $1.399 \pm 0.144$ | $0.945 \pm 0.021$ | $1.167 \pm 0.183$ | $0.789 \pm 0.194$ | 200 | $1.052 \pm 0.209$ | $0.625 \pm 0.109$ |
| | | | 0.05 | $1.226 \pm 0.065$ | $0.911 \pm 0.011$ | $1.158 \pm 0.200$ | $0.795 \pm 0.194$ | | $1.196 \pm 0.716$ | $0.700 \pm 0.124$ |
| TimeFlow | 200 | 0.50 | 0.50 | $0.163 \pm 0.009$ | $0.240 \pm 0.007$ | $0.233 \pm 0.009$ | $\underline{0.230 \pm 0.006}$ | | $0.330 \pm 0.046$ | $0.223 \pm 0.032$ |
| | | 0.30 | 0.30 | $0.331 \pm 0.014$ | $0.396 \pm 0.010$ | $0.419 \pm 0.015$ | $0.370 \pm 0.009$ | 200 | $0.518 \pm 0.057$ | $0.331 \pm 0.038$ |
| | | 0.05 | 0.05 | $0.963 \pm 0.019$ | $0.811 \pm 0.011$ | $1.303 \pm 0.103$ | $0.830 \pm 0.028$ | | $0.877 \pm 0.077$ | $0.707 \pm 0.098$ |
| TV-INRs | 200 | $\sim \mathcal{S}$ | 0.50 | $\mathbf{0.113 \pm 0.018}$ | $\mathbf{0.212 \pm 0.015}$ | $\mathbf{0.188 \pm 0.041}$ | $\mathbf{0.212 \pm 0.027}$ | | $\mathbf{0.038 \pm 0.031}$ | $\mathbf{0.089 \pm 0.035}$ |
| | | | 0.30 | $\mathbf{0.135 \pm 0.027}$ | $\mathbf{0.232 \pm 0.021}$ | $\mathbf{0.214 \pm 0.042}$ | $\mathbf{0.228 \pm 0.028}$ | 200 | $\mathbf{0.051 \pm 0.051}$ | $\mathbf{0.098 \pm 0.042}$ |
| | | | 0.05 | $\mathbf{0.318 \pm 0.063}$ | $\mathbf{0.368 \pm 0.041}$ | $\mathbf{0.453 \pm 0.074}$ | $\mathbf{0.368 \pm 0.042}$ | | $\mathbf{0.244 \pm 0.226}$ | $\mathbf{0.234 \pm 0.099}$ |

For imputation, we compared TV-INRs against the selected baselines across varying signal lengths $L$. We used $L = 2000$ (2K) and $10000$ (10K) time points to match published baseline experiments, and $L = 200$ time points to evaluate performance in lower-data regimes. We define the rate of observed data points during testing as $\tau_{Test}$. The **low-data regime** is characterized by conditions of data scarcity, which includes all scenarios with a limited training set ($L = 200$) and sparse test-time observations $\tau_{Test} \in \{0.5, 0.3, 0.05\}$) as well as the

Table 2: **Univariate forecasting results** with history length $H$, training/testing forecasting lengths $F_{\text{train,test}}$, and MSE/MAE evaluated for forecasting. Bold indicates the best mean performance and underlined values indicate the second-best mean performance (lower is better).

| Model | H | $F_{\text{train}}$ | $F_{\text{test}}$ | Electricity | | Traffic | | Solar-H | |
|---|---|---|---|---|---|---|---|---|---|
| | | | | MSE | MAE | MSE | MAE | MSE | MAE |
| DeepTime | 512 | 96 | 96 | $0.436 \pm 0.020$ | $0.503 \pm 0.016$ | $0.419 \pm 0.103$ | $0.411 \pm 0.047$ | $0.641 \pm 0.183$ | $0.651 \pm 0.089$ |
| | | 192 | 192 | $0.551 \pm 0.157$ | $0.525 \pm 0.055$ | $0.382 \pm 0.056$ | $0.372 \pm 0.027$ | $\mathbf{0.432 \pm 0.121}$ | $0.514 \pm 0.081$ |
| | | 336 | 336 | $\underline{0.793 \pm 0.046}$ | $0.689 \pm 0.037$ | $0.446 \pm 0.107$ | $0.397 \pm 0.058$ | $0.821 \pm 0.013$ | $0.804 \pm 0.002$ |
| | | 720 | 720 | $10.178 \pm 0.218$ | $0.970 \pm 0.178$ | $0.485 \pm 0.059$ | $0.406 \pm 0.014$ | $0.793 \pm 0.041$ | $0.741 \pm 0.001$ |
| TimeFlow | 512 | 96 | 96 | $\underline{0.425 \pm 0.057}$ | $\underline{0.318 \pm 0.050}$ | $\mathbf{0.289 \pm 0.113}$ | $\mathbf{0.281 \pm 0.064}$ | $\underline{0.503 \pm 0.424}$ | $\underline{0.336 \pm 0.142}$ |
| | | 192 | 192 | $\underline{0.498 \pm 0.078}$ | $\mathbf{0.362 \pm 0.060}$ | $\mathbf{0.324 \pm 0.076}$ | $\mathbf{0.298 \pm 0.050}$ | $\underline{0.476 \pm 0.191}$ | $\mathbf{0.352 \pm 0.077}$ |
| | | 336 | 336 | $1.347 \pm 0.210$ | $\mathbf{0.389 \pm 0.065}$ | $\underline{0.407 \pm 0.122}$ | $\underline{0.329 \pm 0.057}$ | $\mathbf{0.364 \pm 0.106}$ | $\mathbf{0.301 \pm 0.055}$ |
| | | 720 | 720 | $\mathbf{9.422 \pm 0.217}$ | $\mathbf{0.525 \pm 0.150}$ | $0.413 \pm 0.050$ | $\underline{0.327 \pm 0.020}$ | $\mathbf{0.353 \pm 0.092}$ | $\mathbf{0.325 \pm 0.032}$ |
| TV-INRs | 512 | $\sim \mathcal{F}$ | 96 | $\mathbf{0.336 \pm 0.068}$ | $\mathbf{0.296 \pm 0.040}$ | $\underline{0.383 \pm 0.143}$ | $\underline{0.305 \pm 0.082}$ | $\mathbf{0.346 \pm 0.303}$ | $\mathbf{0.325 \pm 0.123}$ |
| | | | 192 | $\mathbf{0.446 \pm 0.107}$ | $\underline{0.415 \pm 0.036}$ | $\underline{0.377 \pm 0.094}$ | $\mathbf{0.294 \pm 0.056}$ | $0.469 \pm 0.125$ | $\underline{0.389 \pm 0.031}$ |
| | | | 336 | $\mathbf{0.544 \pm 0.216}$ | $\underline{0.442 \pm 0.040}$ | $\mathbf{0.373 \pm 0.073}$ | $\mathbf{0.292 \pm 0.049}$ | $\underline{0.451 \pm 0.140}$ | $\underline{0.383 \pm 0.039}$ |
| | | | 720 | $\underline{9.515 \pm 0.218}$ | $\underline{0.535 \pm 0.162}$ | $\underline{0.448 \pm 0.088}$ | $\mathbf{0.313 \pm 0.043}$ | $\underline{0.509 \pm 0.194}$ | $\underline{0.404 \pm 0.061}$ |

experiments with a larger training set but very sparse test-time observations ($L = 2000$, $\tau_{Test} = 0.05$). In contrast, **the high-data regime** represents scenarios with a relative abundance of data, specifically when a larger training set is available ($L = 2000$) and the observation rates at test time are higher ($\tau_{Test} \in \{0.5, 0.3\}$) or when $L = 10000$ and $\tau_{Test} \in \{0.5, 0.3, 0.05\}$. To improve robustness under low observation rates, we sample the observed fraction at random during training, e.g. $\tau_{\text{Train}} \sim S = \{0.05, 0.30, 0.50, 0.75, 0.90, 1.0\}$. TimeFlow requires separate training for each $\tau_{Test}$ value, while SAITS fixes $\tau_{\text{Train}} = 0.80$ and CSDI uses a uniform distribution $\tau_{\text{Train}} \sim \mathcal{U}(0, 1)$.

The results in Table 1 demonstrate the advantages of our approach over gradient-based meta-learning, particularly in low-data regimes. With shorter signals ($L = 200$) and lower observation percentages $\tau_{\text{Test}}$, TV-INRs consistently performs on par or better than all baselines, achieving up to 88% improvement in MSE scores. In Solar-10 at ($L = 200$) specifically, TV-INRs achieves substantially lower error rates, with a MSE of 0.038 compared to TimeFlow's 0.330, SAITS' 0.066 and CSDI's 1.010 at $\tau_{\text{Test}} = 0.50$. At the highest missingness setting, $\tau_{\text{Test}} = 0.05$, TV-INRs also performs best on average, though it is only comparable to TimeFlow on the Solar-10 dataset. As Solar-10 has significantly longer time series ($L = 10K$) and thus a larger number of training observations, results indicate that TV-INRs excels primarily in low-data regimes.

For longer signal lengths ($L = 2K, 10K$) and higher observation rates $\tau_{\text{Test}}$, TimeFlow demonstrates stronger performance on the Electricity dataset, while SAITS performs better on the Traffic dataset. Overall, TV-INRs *maintains competitive performance across all scenarios while offering two crucial advantages: it provides a unified model that handles all cases without requiring per-case training, and enables efficient inference through gradient-free meta-learning that requires only a forward pass.* These results highlight how our variational framework effectively balances performance with practical efficiency, and excels in scenarios where data availability is limited. In App. B.6, Figures 5-6 show sample outputs generated by TV-INRs.

### 4.1.2 Forecasting on univariate datasets

For forecasting, we compare TV-INRs with TimeFlow and DeepTime using the same experimental settings as in their original publications, e.g. by modeling each series independently for univariate datasets. The historical length $H$ is set to the first 512 elements, and forecasting performance is evaluated over forecasting lengths $F$ of 96, 192, 336, and 720. TV-INRs is trained by sampling forecasting lengths $F_{\text{Train}} \in \mathcal{F} = \{96, 192, 336, 720\}$. Since $H$ is fixed, the binary mask has the same number of observed indices; however, the total length of the mask is adapted to different lengths of $F$. As shown in Table 2, both TimeFlow and DeepTime require separate training for each forecasting length, while our approach uses a single model for all horizons. For TV-INRs and TimeFlow, there is a dramatic increase in MSE for long-range forecasting ($F = 720$) in the Electricity dataset, reaching $\approx 9.5$ and $\approx 9.4$ respectively, while maintaining relatively moderate MAE ($\approx 0.53$), which strongly indicates the presence of significant outlier errors in the predictions, we further investigate it in App. B.2. DeepTime shows even higher errors in this scenario (MSE = 10.18). For the

other datasets and forecasting lengths, our method and TimeFlow have similar performance, with our model performing better on average for the shortest forecasting horizons ($F = \{96\}$), and TimeFlow performing better for the longest horizons ($F = \{720\}$). Both methods outperform DeepTime on the majority of datasets and horizon lengths. It should be noted that TimeFlow requires separate training per horizon and gradient-based meta-learning for each test sample. Similarly, DeepTime needs individual models for each forecast length. These results demonstrate that, *with a single trained model, TV-INRs handles multiple forecasting horizons while maintaining competitive performance to TimeFlow and outperforming DeepTime.* Sample predictions from TV-INRs are shown in App. B.6 (Fig.7). We further stress-test forecasting under partially observed history; the full protocol and results are deferred to Appendix B.3.

### 4.1.3 Model generalization and complexity

We assess model generalization by its robust performance across a range of distinct tasks, each applied to $N$ unique time series. For imputation, these tasks are defined by varying the observation rate $\tau$, challenging the model under different levels of data scarcity. For forecasting, we measure generalization by the model's ability to maintain accuracy over increasingly long forecasting windows, $\mathcal{F} \in \{96, 192, 336, 720\}$. TV-INRs uses a unified model capable of imputation with different observed ratios and forecasting across all horizon lengths, which significantly reduces or eliminates the need for additional fine-tuning or multiple-model optimizations, enhancing its overall efficiency. To illustrate this, we show that TimeFlow has to be trained per scenario, e.g. different observed ratios and horizon lengths, in Table 20 in App.A.6. We report the training times for TV-INRs and TimeFlow across all experiments in App. A.10. Our findings indicate that TV-INRs achieves notable improvements in cumulative training efficiency: it requires between $2.41\times$ to $3.70\times$ less training time than TimeFlow for forecasting tasks, and between $1.30\times$ to $2.81\times$ less training time for imputation tasks. These results are shown in App. A.10 - Table 21, and demonstrate that TV-INRs offers substantial advantages in computational efficiency and generalization by handling multiple tasks with a single training. We also provide the memory and time complexity analysis of TV-INR in App. A.9.

TV-INRs introduce additional architectural components compared to simpler baselines, including Transformer-based encoders, a hypernetwork, and a variational latent space, resulting in higher architectural complexity. A detailed time and memory complexity analysis is provided in App A.9, where we show that the Transformer-based encoder is the dominant computational bottleneck, with time and memory complexity scaling as $\mathcal{O}(NL^2E)$ and $\mathcal{O}(ML^2)$, respectively.

Importantly, these costs are mainly prominent during training and are amortized across all downstream tasks. At deployment, TV-INRs performs inference via a single forward pass, without per-sample optimization, yielding fixed inference latency (App A.11). In contrast, gradient-based meta-learning approaches such as TimeFlow require iterative test-time optimization, causing inference and cumulative training costs to scale with the number of tasks and adaptation steps (App A.10). Overall, TV-INRs trades higher per-model complexity for amortized efficiency, unified deployment, and predictable inference costs.

### 4.1.4 Imputation on multivariate datasets

**In the HAR dataset,** motion data from a single smartphone presents simultaneous missing values across all channels at specific timestamps due to device failures. Formally, given $\boldsymbol{X}^{(i)} = \boldsymbol{X}^{(i)}_{\text{obs}} \cup \boldsymbol{X}^{(i)}_{\text{unobs}}$, where $\boldsymbol{X}^{(i)}_{\text{unobs}} = \boldsymbol{X}^{(i)}_l : l \in \mathcal{U}^{(i)}$, any missing timestamp $l \in (\mathcal{U}^{(i)})$ affects all $d$ channels.

**For the P12 dataset,** we evaluate TV-INRs on patient-specific time series imputation from eight measurements (urine output, SysABP, DiasABP, MAP, HR, NISysABP, NIDiasABP, NIMAP) and four covariates (gender, age, height, weight). The dataset has irregular missingness across timestamps and channels, which makes the imputation task more challenging (details in App. A.2).

- **Conditional vs. unconditional.** We test C-TV-INRs conditional formulation (Equation 2) on HAR by incorporating activity labels alongside latent codes, and on P12 by including patient demographic information (age, gender, height, weight). On HAR, Table 3 shows C-TV-INRs outperforms TV-INRs at all missingness rates, and both TV-INRs models outperform SAITs and CSDI. For P12, the non-conditional version of our model is best at the highest observation rate ($\tau_{\text{Test}} = 0.50$), while the conditional version outperforms it under moderate and extreme sparsity ($\tau_{\text{Test}} = 0.30, 0.10$). These results suggest that TV-INRs is superior for multivariate imputation on these datasets, and the conditional model is particularly useful for sparse time series.

Table 3: **Multivariate imputation results** with signal lengths $L$, training/testing observation rates $\tau_{\text{train,test}}$, and MSE/MAE evaluated on unobserved indices from non-overlapping test signals. Bold indicates the best mean performance and underlined values indicate the second-best mean performance (lower is better).

| Model | | | HAR (L=128) | | | | P12 (L=48) | |
|---|---|---|---|---|---|---|---|---|
| | $\tau_{\text{Train}}$ | $\tau_{\text{Test}}$ | MSE | MAE | $\tau_{\text{Train}}$ | $\tau_{\text{Test}}$ | MSE | MAE |
| SAITS | 0.80 | 0.50 | $0.998 \pm 0.003$ | $0.793 \pm 0.006$ | 0.80 | 0.50 | $0.985 \pm 0.128$ | $0.746 \pm 0.070$ |
| | | 0.30 | $1.001 \pm 0.004$ | $0.793 \pm 0.007$ | | 0.30 | $0.998 \pm 0.092$ | $0.760 \pm 0.067$ |
| | | 0.05 | $1.004 \pm 0.001$ | $0.793 \pm 0.007$ | | 0.10 | $\underline{0.970 \pm 0.048}$ | $0.746 \pm 0.052$ |
| CSDI | $\sim \mathcal{U}$ | 0.50 | $1.083 \pm 0.062$ | $0.821 \pm 0.067$ | $\sim \mathcal{U}$ | 0.50 | $0.861 \pm 0.174$ | $0.691 \pm 0.070$ |
| | | 0.30 | $1.084 \pm 0.060$ | $0.823 \pm 0.063$ | | 0.30 | $0.930 \pm 0.146$ | $0.724 \pm 0.067$ |
| | | 0.05 | $1.090 \pm 0.015$ | $0.826 \pm 0.054$ | | 0.10 | $1.024 \pm 0.093$ | $0.765 \pm 0.057$ |
| TV-INRs | $\sim \mathcal{S}$ | 0.50 | $\underline{0.382 \pm 0.067}$ | $\underline{0.414 \pm 0.041}$ | $\sim \mathcal{S}$ | 0.50 | $\mathbf{0.822 \pm 0.171}$ | $\mathbf{0.660 \pm 0.074}$ |
| | | 0.30 | $\underline{0.533 \pm 0.050}$ | $\underline{0.505 \pm 0.031}$ | | 0.30 | $\underline{0.892 \pm 0.146}$ | $\underline{0.692 \pm 0.071}$ |
| | | 0.05 | $\underline{0.995 \pm 0.070}$ | $\underline{0.722 \pm 0.034}$ | | 0.10 | $0.980 \pm 0.118$ | $\underline{0.739 \pm 0.058}$ |
| C-TV-INRs | $\sim \mathcal{S}$ | 0.50 | $\mathbf{0.379 \pm 0.065}$ | $\mathbf{0.412 \pm 0.041}$ | $\sim \mathcal{S}$ | 0.50 | $\underline{0.824 \pm 0.175}$ | $\underline{0.662 \pm 0.076}$ |
| | | 0.30 | $\mathbf{0.523 \pm 0.047}$ | $\mathbf{0.502 \pm 0.029}$ | | 0.30 | $\mathbf{0.883 \pm 0.141}$ | $\mathbf{0.690 \pm 0.073}$ |
| | | 0.05 | $\mathbf{0.976 \pm 0.058}$ | $\mathbf{0.708 \pm 0.022}$ | | 0.10 | $\mathbf{0.963 \pm 0.099}$ | $\mathbf{0.733 \pm 0.052}$ |

- **Downstream classification.** To assess the impact of imputation on classification, we trained an XGBoost classifier (Chen & Guestrin, 2016) on HAR data, testing across varying observation ratios by removing random timepoints and imputing using our methods, baselines, and mean imputation. Fig. 3 shows both TV-INRs variants outperforming baselines, with the conditional model showing increasing advantage as missingness grows, demonstrating the value of covariates for individualized predictions. Complete AUC-ROC values are in Table 12.

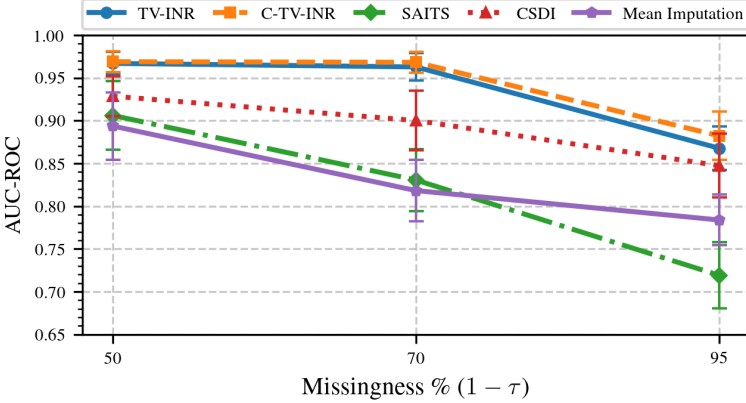

Figure 3: Classification performance (AUC-ROC) at various missingness levels; a higher value indicates better performance.

### 4.1.5 Forecasting on multivariate datasets

Here we add the results for multivariate forecasting. In this setting, each dataset (Electricity, Traffic, and Solar-H) is modeled as a multivariate signal by jointly processing multiple independent time series. This task is inherently more challenging for TV-INRs compared to the per-series univariate setup, as the model must learn shared latent structure across heterogeneous time-series rather than specializing to each series individually. In contrast to our univariate experiments, where adaptation occurs at the signal level, the multivariate formulation requires modeling independent dynamics jointly, which increases representational burden and reduces implicit per-series flexibility. For fair comparison with DeepTime, we follow their experimental protocol and evaluate aggregated multivariate forecasting performance. We exclude TimeFlow from this comparison, as its current implementation is restricted to the univariate case and does not support joint multivariate modeling.

Table 4: **Multivariate forecasting results** with history length $H$, training/testing forecasting lengths $F_{\text{train,test}}$, and MSE/MAE evaluated for forecasting.

| Model | H | $F_{\text{train}}$ | $F_{\text{test}}$ | Electricity | | Traffic | | Solar-H | |
|---|---|---|---|---|---|---|---|---|---|
| | | | | MSE | MAE | MSE | MAE | MSE | MAE |
| DeepTime | 512 | 96 | 96 | **0.667 ± 0.444** | 0.642 ± 0.224 | **0.690 ± 0.211** | 0.574 ± 0.977 | 1.021 ± 0.152 | 0.874 ± 0.407 |
| | | 192 | 192 | **0.864 ± 1.816** | 0.728 ± 0.747 | **0.775 ± 2.024** | 0.605 ± 0.717 | 0.812 ± 0.240 | 0.735 ± 0.212 |
| | | 336 | 336 | **1.093 ± 0.132** | **0.860 ± 0.488** | **0.985 ± 0.149** | 0.715 ± 0.486 | 0.918 ± 0.117 | 0.790 ± 0.602 |
| | | 720 | 720 | 10.440 ± 2.185 | **0.974 ± 0.182** | **1.106 ± 0.116** | **0.759 ± 0.252** | 0.894 ± 0.891 | 0.713 ± 0.788 |
| TV-INRs | 512 | $\sim \mathcal{F}$ | 96 | 0.684 ± 0.390 | **0.601 ± 0.205** | 0.702 ± 0.205 | **0.561 ± 0.880** | 1.009 ± 0.148 | 0.853 ± 0.382 |
| | | | 192 | 0.882 ± 0.980 | **0.694 ± 0.620** | 0.801 ± 1.900 | **0.598 ± 0.701** | 0.790 ± 0.264 | 0.722 ± 0.204 |
| | | | 336 | 1.176 ± 0.140 | 0.889 ± 0.470 | 1.034 ± 0.160 | 0.738 ± 0.470 | 0.956 ± 0.130 | 0.812 ± 0.610 |
| | | | 720 | **10.130 ± 2.200** | 1.012 ± 0.210 | 1.248 ± 0.130 | 0.804 ± 0.260 | 0.948 ± 0.910 | 0.736 ± 0.820 |

To accommodate the increased complexity of the multivariate forecasting task, we scale the capacity of TV-INRs accordingly. Specifically, we increase the latent dimensionality to $d_z = 256$, expand the INR decoder to five layers with inner dimension 128, and enlarge the hypernetwork to inner dimension 256. The Transformer encoder is strengthened to $d_{\text{model}} = 256$ with four attention heads. Additionally, we increase the number of Fourier feature frequencies to 512 to better capture cross-series temporal variation. These modifications ensure that TV-INRs has sufficient representational power to handle the multivariate forecasting setting while maintaining the same amortized inference framework without per-horizon or per-series retraining.

Table 4 reports multivariate forecasting results under the joint modeling setting. TV-INRs achieves competitive performance at shorter horizons ($F = 96, 192$), often improving MAE while remaining close in MSE, indicating accurate central tendency estimation without horizon-specific tuning. At longer horizons ($F = 336, 720$), DeepTime generally attains lower MSE, reflecting the advantage of per-horizon optimization. Notably, in the Electricity dataset at $F = 720$, both DeepTime and TV-INRs exhibit substantial errors in MSE, consistent with the long-horizon instability observed in the univariate forecasting setting.

### 4.1.6 Failure Modes and Limitations

While TV-INRs demonstrates strong performance across a wide range of imputation and forecasting settings, we identify two systematic limitations.

**High-data regimes.** When long sequences are densely observed (i.e., large $L$ with high $\tau_{\text{test}}$), gradient-based meta-learning approaches such as TimeFlow can outperform TV-INRs, particularly on univariate datasets such as Electricity and Traffic (Table 1). This behavior is consistent with prior observations that per-instance optimization can better exploit abundant data to finely adapt INR parameters, whereas amortized inference prioritizes robustness across different regimes. Notably, this performance gap primarily arises in scenarios where retraining or per-sample optimization is computationally feasible. Overall, these findings highlight an inherent trade-off between amortized, scenario-agnostic inference and task-specific optimization, positioning TV-INRs as particularly well-suited for sparse, irregular, or deployment-constrained settings rather than fully observed, high-data regimes. To mitigate this limitation, long sequences with highly heterogeneous temporal structure may benefit from hierarchical or segment-wise latent variables instead of a single global latent vector, enabling finer-grained adaptation while preserving amortized inference.

**Long-horizon forecasting instability.** Similar to competing continuous-time and discrete-time models, TV-INRs exhibits performance degradation for very long forecasting horizons (e.g., $F = 720$), with occasional large errors reflected in elevated MSE despite moderate MAE (Table 2). This suggests sensitivity to rare but extreme prediction failures. We emphasize that this behavior is not unique to TV-INRs and is also observed in DeepTime under identical conditions.

**Missingness representation** TV-INRs represents missingness with zero-filled unobserved values and a binary indicator mask. This mask is also explicitly used in the attention mechanism to prevent unobserved entries from influencing the encoder, thereby prioritizing information from observed values while maintaining a consistent input structure. However, this formulation treats missingness as structural rather than potentially informative. In some domains (e.g., healthcare), missing observations may carry semantic meaning, and excluding them from attention may limit the model's ability to exploit such signals. While temporal information

is provided separately through continuous coordinates encoded via Fourier features, richer missingness-aware representations or explicit modeling of observation processes could introduce beneficial inductive biases. We consider this a promising direction for future work.

## 5 Conclusion

We have introduced TV-INRs, demonstrating its effectiveness in imputation and forecasting across various time series domains and data conditions. Our results highlight superior performance in low-data regimes and robust handling of varying observation patterns, in comparison to state-of-the-art INR time series models and several additional baselines. Furthermore, the amortization of INR weights in our probabilistic setting enables adaptation to unseen data without fine-tuning or per-sample optimization, a key advantage over traditional hypernetwork-based methods that rely on meta-learning. We have also illustrated the potential of TV-INRs for downstream tasks with improved classification on HAR data. While baseline methods TimeFlow and DeepTime showed stronger performance in specific scenarios, TV-INRs frequently produced comparable or superior results while offering substantial practical benefits: unified model training across multiple tasks, individualization without meta-learning, significantly improved cumulative training and fixed inference time, independent of gradient adaptation. The ability to handle multiple forecasting horizons with a single model represents a considerable advantage in real-world applications where computational resources may be limited.

To further enhance our model, future directions may include reducing hypernetwork complexity with transformer-based architectures (Chen & Wang, 2022), modeling per-sample positional embeddings rather than weights directly (Park et al., 2024), and possibly extending TV-INRs toward explicit out-of-domain generalization in foundation-model settings (Naour et al., 2025).

The variational framework could also be extended to incorporate additional forms of domain knowledge. These improvements could strengthen its potential, particularly in healthcare domains such as personalized medicine and patient monitoring, where efficiency and the ability to model highly sparse data are especially critical.

## 6 Broader Impact

This paper presents work that aims to increase the efficiency and scalability of generative models in Machine Learning. There are many potential societal consequences of our work, none which we feel must be specifically highlighted here.

## 7 Acknowledgments

This work has been supported by the project *"Society-Aware Machine Learning: The paradigm shift demanded by society to trust machine learning"*, funded by the European Union and led by Isabel Valera (ERC-2021-STG, SAML, 101040177). Views and opinions expressed are, however, those of the author(s) only and do not necessarily reflect those of the aforementioned funding agencies. Neither of the aforementioned parties can be held responsible for them. Rachael DeVries and Ole Winther were in part funded by the Novo Nordisk Foundation through the Center for Basic Machine Learning Research in Life Science (NNF20OC0062606). Ole Winther was in part funded by the Novo Nordisk Foundation through CAZAI (NNF22OC0077058). We acknowledge support from the Pioneer Center for AI, DNRF grant number P1.

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

# Appendix

## A   Appendix A

### A.1   Reproducibility Statement

Our work is fully reproducible, and all the necessary resources are provided below.

**Code.** The full implementation of our method, including training and evaluation scripts, will be made publicly available upon publication (anonymous link provided for review `https://anonymous.4open.science/r/TV INR-codebase-8C08`).

**Data.** Instructions for obtaining the raw data are included with the code repository. A detailed description of the datasets, preprocessing, and normalization steps is provided in Appendix-A.2.

**Model and Training.** The model architecture, training protocol, and evaluation procedure are described in Sections 3.2 and 4. Hyperparameter choices and tuning procedures are reported in Appendix A.6.

**Hardware and Software.** All experiments were run on a single NVIDIA V100 GPU. A complete list of dependencies and environment details is provided in our codebase.

### A.2   Datasets

Table 5: **Dataset Descriptions.** #Series denotes the number of distinct timeseries signals with corresponding lenghts and covariates if available.

| Dataset | Domain | Freq. | #Dims | #Series | Length | Cov. |
|---|---|---|---|---|---|---|
| Electricity | $\mathbb{R}_0{}^+$ | Hourly | 1 | 321 | 26304 | ✗ |
| Traffic | $[0,1]$ | Hourly | 1 | 862 | 17544 | ✗ |
| Solar-10 | $\mathbb{R}_0{}^+$ | 10 Mins | 1 | 137 | 52560 | ✗ |
| Solar-H | $\mathbb{R}_0{}^+$ | Hourly | 1 | 137 | 8760 | ✗ |
| HAR | $\mathbb{R}$ | 50Hz | 3 | 30 | 43940 | ✓ |
| P12 | $\mathbb{R}_0{}^+$ | Hourly | 8 | 3938 | 48 | ✓ |

In this section, we provide more details about the datasets we have used. We start with the list of uni-variate datasets:

**Electricity Dataset** records hourly electricity consumption from 321 customers in Portugal for the period 2012 to 2014, displaying both daily and weekly seasonality.

**Traffic Dataset** includes hourly road occupancy rates from 862 locations in San Francisco during 2015 and 2016, and exhibits similar daily and weekly seasonal patterns.

**Solar Dataset** The Solar-10 dataset comprises measurements of solar power production from 137 photovoltaic plants in Alabama, captured every 10 minutes in 2006. Additionally, there is an hourly version of this dataset, known as Solar-Hourly.

For some datasets, the feature vectors $\boldsymbol{Y}^{(i)} = \{\mathbf{y}_l^{(i)}\}_{l=1}^{L_i}$ expand from univariate ($d = 1$) to multivariate ($d > 1$), with each dimension representing a unique sensor used to collect observations $\{\mathbf{y}_l^{(i)}\} \in \mathbb{R}^d$. For these purposes, we experiment with two multi-variate datasets, namely:

**HAR Dataset.** Here, we experiment with the Human Activity Recognition (HAR) dataset from the UC Irvine ML Repository, which is dense with regular time points at 2.56 second intervals, enabling quantitative imputation assessment through random removal. It contains 10,299 samples of accelerometer measurements across x, y, and z axes.

**P12 Dataset.** The PhysioNet Challenge 2012 (P12) dataset contains ICU stay measurements including sensor readings and lab results. After outlier removal, it comprises 11,817 visits across 37 channels with maximum 215 time points over 48 hours. We use eight measurements urine output, systolic arterial blood pressure

(SysABP), diastolic arterial blood pressure (DiasABP), mean arterial pressure (MAP), heart rate (HR), and their non-invasive counterparts (NISysABP, NIDiasABP, NIMAP). We also incorporate patient-specific covariates including gender, age, height, and weight. Conditional TV-INRs use covariates Unlike HAR, P12 is highly sparse ($\boldsymbol{X}_{\text{obs}}^{(i)}$ is 15.68% of $\boldsymbol{X}$ on average) with irregularity across times and sensors, where $\boldsymbol{T}^{(i)}$ may be unique for each time series $i$.

**Missingness Patterns of the Datasets.** To ensure a comprehensive evaluation, our experiments address diverse data missingness patterns, including both random and non-random scenarios. For Missing Completely at Random (MCAR) patterns, we adhere to standard literature practices by introducing artificial missingness (Little & Rubin, 2019) during training across the Electricity, Traffic, and Solar datasets. This methodology aligns with the protocols used by the baseline models we compare against. Furthermore, we assess performance on Missing Not at Random (MNAR) patterns, which are prevalent in real-world applications. Our analysis includes the P12 dataset, which exhibits MNAR characteristics where clinical data is informatively missing; here, we evaluate imputation quality indirectly via a downstream classification task. To create a controlled non-random evaluation, we also synthetically modified the fully-observed HAR dataset by dropping entire channels at random timestamps to mimic sensor failures, a scenario where the missingness mechanism depends on unobserved factors.

## A.3 Data-preprocessing

We apply channel-wise standardization to each time series. For each channel $d$ in a time series with length $L$, we compute the channel-wise mean $\mu_d$, standard deviation $\sigma_c$, and normalize signal $\hat{x}_{l,d}^{(i)}$ as follows:

$$\hat{x}_{l,d}^{(i)} = \frac{x_{l,d}^{(i)} - \mu_d^{(i)}}{\sigma_d^{(i)}} \tag{9}$$

where $x_{l,d}^{(i)}$ represents the value of channel $d$ at time $l$ for sample $i$.

## A.4 Analysis for statistical differences

To compare the performance of TV-INRs and baseline models, we conducted a systematic statistical analysis using Welch's t-test which accounts for potentially unequal variances between the two models. For each configuration defined by sequence length $L$ and sampling ratio $\tau$, we evaluated both mean squared error (MSE) and mean absolute error (MAE). The statistical significance was assessed at $\alpha = 0.05$.

In classification experiments, the HAR dataset was normalized independently per channel but not per individual, ensuring consistency across subjects and allowing XGBoost to learn global patterns. This differs from the normalization procedure used for TV-INRs, which normalized data at both the channel and individual level in order to model data on a per-user basis. When mentioned, we computed the relative performance difference as $\Delta = (\mu_{\text{TimeFlow}} - \mu_{\text{TV-INRs}})/\mu_{\text{TimeFlow}} \times 100\%$.

## A.5 Training, validation, and test splits for all experiments

Here, we give information about all datasplits for all experiments in Tables 6, 7, 8. For univariate datasets, test windows are extracted sequentially from the end of each time series. Moreover, training data precedes validation data.

---

[1]NO: Non-overlapping, FE: From end of the series

Table 6: Dataset splitting details for univariate imputation experiments. Training and validation sets has 5:1 ratio.

| Dataset | Series Count | Window Length (L) | Test Windows (NO & FE )[1] | Training/Val. Stride |
|---|---|---|---|---|
| Electricity | 321 | 200
2000 | 50
5 | 50
500 |
| Traffic | 862 | 200
2000 | 20
2 | 50
500 |
| Solar-10 | 137 | 200
10000 | 100
2 | 50
250 |

Table 7: Dataset splitting details for univariate forecasting experiments. Training and validation sets has 5:1 ratio. Training and validation series are constructed with using offsetting from the available data points.

| Dataset | Series Count | History (H) | Forecast (F) | Window Length (L) | Test Windows (NO & FE )[2] | Training/Val. Offset |
|---|---|---|---|---|---|---|
| Electricity | 321 | 512 | [96,192,336,720] | 1232 | 7 | ✓ |
| Traffic | 862 | 512 | [96,192,336,720] | 1232 | 7 | ✓ |
| Solar-H | 137 | 512 | [96,192,336,720] | 1232 | 3 | ✓ |

Table 8: Dataset splitting details for HAR imputation experiments. The dataset is split by users, with 24 users for training and 6 users for testing. From the training users, we further split into training and validation sets using a 4:1 ratio of users.

| Dataset | Series Count | Window Length (L) | #Classes | #Train Users | #Test Users |
|---|---|---|---|---|---|
| HAR | 30 | 128 | 6 | 24 | 6 |
| P12 | 11817 | 48 | NA | 9454 | 2363 |

## A.6 Hyperparameters for all experiments

Hyperparameters for all TV-INR experiments on an NVIDIA V100 GPU can be seen in Tables 9-10. In case of HAR dataset, C-TV-INRs extra parameters of feed forward encoder of covariates with layers $[8, 8]$ and dim_c $= 4$. The details of the hyperparameter grid search space are provided in Table 11.

Table 9: Hyperparameter details of TV-INRs for imputation task.

| | | ELECTRICITY | | TRAFFIC | | SOLAR-10 | | HAR |
|---|---|---|---|---|---|---|---|---|
| | L | 200 | 2000 | 200 | 2000 | 200 | 10000 | 128 |
| | dim_z | 32 | 64 | 32 | 64 | 32 | 64 | 32 |
| | epochs | 2000 | 4000 | 2000 | 4000 | 2000 | 4000 | 3000 |
| | bs | 256 | 64 | 256 | 64 | 256 | 32 | 128 |
| | lr | 1e-4 | 1e-4 | 1e-4 | 1e-4 | 1e-4 | 1e-4 | 1e-4 |
| | $d_{model}$ | 128 | 128 | 128 | 128 | 128 | 128 | 128 |
| Transformer Enc. | #heads | 2 | 4 | 2 | 4 | 2 | 4 | 4 |
| | #layers | 2 | 2 | 2 | 2 | 2 | 2 | 4 |
| Hypernetwork | layers | [128,256] | | | | | | |
| Generator | layers | [64,64,64] | [64,64,64,64] | [64,64,64] | [64,64,64,64] | [64,64,64] | [64,64,64,64] | [64,64,64,64] |
| RFF | | m $= 256, \sigma = 2$ | | | | | | |

Table 10: Hyperparameter details of TV-INRs for forecasting task.

|  |  | ELECTRICITY | TRAFFIC | SOLAR-H |
|---|---|---|---|---|
|  | dim_z | 32 | 64 | 32 |
|  | max epochs | 2000 | 4000 | 2000 |
|  | bs | 256 | 64 | 256 |
|  | lr | 1e-4 | 1e-4 | 1e-4 |
| Transformer Enc. | $d_{model}$ | 128 | 128 | 128 |
|  | #heads | 2 | 4 | 2 |
|  | #layers | 2 | 2 | 2 |
| Hypernetwork | layers | [128,256] | | |
| Generator | layers | [64,64,64] | [64,64,64,64] | [64,64,64] |
| Random Fourier Features |  | $m = 256, \sigma = 2$ | | |

Table 11: Hyperparameter Grid Search Configuration

| Hyperparameter | Search Range |
|---|---|
| **General Parameters** | |
| Learning rate (lr) | [1e-5, 1e-4, 5e-4] |
| Latent dimension (dim_z) | [16, 32, 64] |
| Dropout rate | [0.0, 0.1, 0.2] |
| **Transformer Encoder** | |
| d_model | [64, 128, 256] |
| Attention layers | [2, 4, 6] |
| Number of heads | [2, 4, 8] |
| Causal attention | [True, False] |
| **Hypernetwork** | |
| Layers | [[32,64], [64,128], [128,256], [256,512]] |
| Activation | ['relu', 'lrelu_01', 'gelu'] |
| **Generator (INR)** | |
| dim_inner | [32,64,128] |
| num_layers | [2, 3, 4] |
| Activation | ['relu', 'lrelu_01', 'gelu'] |
| **Random Fourier Features** | |
| m | [128, 256, 512] |
| $\sigma$ | [1, 2, 4] |

**For classification with XGBoost**, all hyperparameters used were the default in Chen & Guestrin (2016)'s XGBoost library, with the following exceptions; early stopping was set to 10 rounds, and categorical features were enabled to preserve channel identity as nonordinal.

### A.7   Classifer results

We present the AUC-ROC scores for different models across varying levels of missingness in Table 12, where higher scores indicate better classification performance.

### A.8   TimeFlow results for different missingness rates

To thoroughly demonstrate TV-INRs's capability to handle different missing data scenarios, we conducted extensive experiments by training and testing with various observed ratios ($\tau$), further supporting our claims regarding its efficiency and its ability to serve as a single model for all cases. It is important to note that in the TimeFlow GitHub repository[3], the missing data rate ("`draw_ratio`") can be set as a training argument, with options including $\{0.05, 0.10, 0.20, 0.30, 0.50\}$. Although this may appear to be a hyperparameter choice, it affects the task itself, as the model is optimized for a specific level of missingness.

---

[3]`https://github.com/EtienneLnr/TimeFlow/blob/main/experiments/training/inr_imputation.sh`

Table 12: AUC-ROC scores for different models across varying levels of missingness. Higher scores indicate better performance. All values are rounded to three decimal places.

| Model | 50% Missingness | 70% Missingness | 95% Missingness |
|---|---|---|---|
| C-TV-INR | **0.969 ± 0.012** | **0.968 ± 0.012** | **0.882 ± 0.028** |
| TV-INR | 0.967 ± 0.013 | 0.963 ± 0.016 | 0.868 ± 0.025 |
| SAITS | 0.906 ± 0.040 | 0.831 ± 0.036 | 0.719 ± 0.039 |
| CSDI | 0.928 ± 0.023 | 0.900 ± 0.035 | 0.847 ± 0.037 |
| Mean Imputation | 0.894 ± 0.039 | 0.818 ± 0.036 | 0.784 ± 0.030 |

As shown in Table 13, TimeFlow's performance varies significantly across different training and testing $\tau$ combinations, requiring separate model instances for each scenario, and although we implemented a version of TimeFlow that can be trained using random observed fractions, this has not yet led to improved results. In contrast, TV-INRs has comparable or better performance when compared with Timeflow with a single trained model. These results align with the observation stated in Table 10 of the original TimeFlow paper (Naour et al., 2024) that while higher sampling rates simplify the imputation task, they complicate optimization, making it challenging for the model to generalize effectively across different sparsity levels.

Table 13: **TimeFlow model performance at different training and testing missing ratios ($\tau$), including random sampling from set $\mathcal{S}$.** MSE and MAE metrics are reported for electricity dataset.

| | | | Test $\tau$ | | | | | |
|---|---|---|---|---|---|---|---|---|
| | | | MSE | | | MAE | | |
| Model | $L$ | Train $\tau$ | 0.05 | 0.3 | 0.5 | 0.05 | 0.3 | 0.5 |
| TimeFlow | 2K | 1.00 | 108812.06 | 0.18195 | 0.13066 | 26.16919 | 0.28272 | 0.25284 |
| | | 0.95 | 22579.357 | 0.15164 | **0.1275** | 15.57548 | 0.27184 | 0.24665 |
| | | 0.50 | 56.5905 | **0.14723** | 0.13238 | 1.88119 | **0.26775** | **0.25075** |
| | | 0.30 | 2.58694 | 0.16536 | 0.15019 | 0.85563 | 0.28756 | 0.27291 |
| | | 0.05 | 0.37793 | 0.22935 | 0.21811 | 0.45838 | 0.34629 | 0.33603 |
| | | $\sim \mathcal{S}$ | 0.32549 | 0.16117 | 0.13834 | 0.38618 | 0.26933 | 0.25845 |
| TV-INRs | 2K | $\sim \mathcal{S}$ | **0.2889** | 0.2502 | 0.2491 | **0.3595** | 0.3317 | 0.3311 |
| TimeFlow | 200 | 1.00 | 605909.85 | 7.77814 | 0.44302 | 358.39774 | 1.87872 | 0.49501 |
| | | 0.95 | 2611667.2 | 145.28325 | 0.33257 | 587.75934 | 2.32136 | 0.42111 |
| | | 0.50 | 350.9098 | 0.34692 | 0.16299 | 11.31193 | 0.43012 | 0.23984 |
| | | 0.30 | 18.90844 | 0.32993 | 0.20594 | 2.99975 | 0.39625 | 0.30289 |
| | | 0.05 | 0.96294 | 0.74811 | 0.6934 | 0.81073 | 0.71435 | 0.69580 |
| | | $\sim \mathcal{S}$ | 0.82365 | 0.33733 | 0.16998 | 0.73533 | 0.3999 | 0.255559 |
| TV-INRs | 200 | $\sim \mathcal{S}$ | **0.3175** | **0.1352** | **0.1132** | **0.3681** | **0.2320** | **0.2123** |

## A.9 Complexity analysis for TV-INR and comparison to baselines

This section provides the time and memory complexity analysis for the TV-INR model, broken down by its core components: the Transformer-based encoder and the MLP-based decoder (hypernetwork).

**Notation.** To facilitate the analysis, we define the following notation: $L$ is the input sequence length; $C$ is the number of input channels; $E$ is the embedding dimension; $D_p$ is the hidden dimension of the projection layer; $Z$ is the latent dimension; $N$ and $M$ are the number of layers and attention heads in the encoder, respectively; $N'$ and $D_h$ are the number of layers and hidden dimensions of the hypernetwork; and $R$ is the total flattened dimension of the INR parameters being modeled. Typically, the sequence length is the dominant factor, such that $L \gg E \gg Z$.

**Time complexity.** The overall time complexity is determined by the sum of the model's parts. The Transformer-based encoder has a complexity of $O(N \cdot L^2 \cdot E)$, which is quadratic with respect to the sequence length $L$ due to the self-attention mechanism. The subsequent projection layer has a complexity of $O(E \cdot D_p)$. The MLP-based hypernetwork's complexity is $O(Z \cdot D_h + (N' - 1) \cdot D_h^2 + D_h \cdot R)$, which depends on its depth

and width. Given that $L$ is the largest dimension, the encoder is the computational bottleneck, making the model's overall time complexity $O(N \cdot L^2 \cdot E)$.

**Memory complexity.** The memory complexity during a forward pass is also dominated by the encoder. The Transformer requires $O(M \cdot L^2)$ memory to store the attention score matrix. The memory requirements for the projection layer and the MLP-based hypernetwork are $O(\max(E, Z))$ and $O(\max(Z, D_h, R))$, respectively, as they are determined by the largest linear layer within each component. Consequently, the overall memory complexity is dictated by the encoder, resulting in $O(M \cdot L^2)$.

**Comparison to SAITS, CSDI, TimeFlow, and DeepTime.** SAITS shares the same $\mathcal{O}(L^2)$ self-attention scaling as TV-INRs due to its Transformer backbone, but operates strictly in discrete time and does not model distributions over continuous generator functions. While architecturally simpler, SAITS requires task-specific configuration and lacks latent-function amortization across varying sparsity levels. CSDI relies on an iterative diffusion process, where both training and inference scale with the number of denoising steps, leading to increased computational cost and non-constant inference latency. Although diffusion provides probabilistic modeling, it operates on fixed temporal grids and requires sequential refinement at test time.

TimeFlow, while also INR-based, performs gradient-based meta-learning and requires separate training for different sparsity levels or forecasting horizons; additionally, inference involves iterative gradient updates per test instance, causing cumulative computational cost to scale with the number of tasks and adaptation steps. DeepTime is computationally lightweight and fast to train, but requires separate models per forecasting horizon and operates in discrete time without continuous-function modeling or latent amortization.

In contrast, TV-INRs performs fully amortized inference via a single forward pass, maintains fixed deployment latency, models continuous-time signals, and unifies multiple sparsity and forecasting regimes within one trained model. Thus, TV-INRs trades higher architectural complexity for amortized efficiency, unified deployment, and predictable inference cost.

### A.10 Training times comparison

In this part, we are reporting the cumulative training times in hours (h) of TV-INRs and Timeflow per task. All training times are rounded to 5-minute intervals and were acquired using an NVIDIA V100 GPU and reported in Tables 14,15,16 and 18,19,20 for imputation and forecasting tasks, respectively. As training times of C-TV-INRs are in the same order with TV-INRs, we omit them to include them in the tables. SAITS demonstrates moderate training times ranging from 1h45m to 13h35m across various datasets, offering a reasonable compromise between efficiency and performance. A drawback of CSDI (Tashiro et al., 2021) is its extended training duration, primarily due to the iterative optimization process inherent in diffusion model training. DeepTime (Woo et al., 2023) is very fast to train due to number of epochs selected in the original work; however it also has the worst performance among the baselines as shown in Table 2. Our primary baseline, TimeFlow, demands significantly greater computational resources, with cumulative training durations consistently exceeding those of TV-INR across most experimental scenarios. Efficiency analyses reveal TimeFlow requires up to 3.70× longer training periods, particularly pronounced in forecasting applications as shown in Table 21.

Table 14: Training times for imputation task, TV-INRs.

| Model Name | Dataset | L | Max Epochs | Training Time |
|---|---|---|---|---|
| TV-INR | Electricity | 200 | 2000 | 8h45m |
| TV-INR | Electricity | 2000 | 4000 | 12h55m |
| TV-INR | Traffic | 200 | 2000 | 10h35m |
| TV-INR | Traffic | 2000 | 4000 | 15h50m |
| TV-INR | Solar-10 | 200 | 2000 | 10h25m |
| TV-INR | Solar-10 | 10000 | 4000 | 19h15m |
| TV-INR | HAR | 128 | 3000 | 6h45m |
| TV-INR | P12 | 128 | 1000 | 4h05m |

Table 15: Training times for imputation task, TimeFlow.

| Model Name | Dataset | L | $\tau$ | Max Epochs | Training Time |
|---|---|---|---|---|---|
| TimeFlow | Electricity | 200 | 0.05 | 40000 | 6h35m |
| TimeFlow | Electricity | 200 | 0.30 | 40000 | 6h40m |
| TimeFlow | Electricity | 200 | 0.50 | 40000 | 6h35m |
| TimeFlow | Electricity | 2000 | 0.05 | 40000 | 5h35m |
| TimeFlow | Electricity | 2000 | 0.30 | 40000 | 5h30m |
| TimeFlow | Electricity | 2000 | 0.50 | 40000 | 5h40m |
| TimeFlow | Traffic | 200 | 0.05 | 40000 | 9h45m |
| TimeFlow | Traffic | 200 | 0.30 | 40000 | 9h50m |
| TimeFlow | Traffic | 200 | 0.50 | 40000 | 10h10m |
| TimeFlow | Traffic | 2000 | 0.05 | 40000 | 8h30m |
| TimeFlow | Traffic | 2000 | 0.30 | 40000 | 8h30m |
| TimeFlow | Traffic | 2000 | 0.50 | 40000 | 8h45m |
| TimeFlow | Solar-10 | 200 | 0.05 | 40000 | 6h45m |
| TimeFlow | Solar-10 | 200 | 0.30 | 40000 | 6h30m |
| TimeFlow | Solar-10 | 200 | 0.50 | 40000 | 6h35m |
| TimeFlow | Solar-10 | 10000 | 0.05 | 40000 | 12h5m |
| TimeFlow | Solar-10 | 10000 | 0.30 | 40000 | 11h50m |
| TimeFlow | Solar-10 | 10000 | 0.50 | 40000 | 12h15m |

Table 16: Training times for imputation task, SAITS.

| Model Name | Dataset | L | Max Epochs | Training Time |
|---|---|---|---|---|
| SAITS | Electricity | 200 | 10000 | 3h45m |
| SAITS | Electricity | 2000 | 10000 | 3h35m |
| SAITS | Traffic | 200 | 10000 | 3h25m |
| SAITS | Traffic | 2000 | 10000 | 7h45m |
| SAITS | Solar-10 | 200 | 10000 | 1h45m |
| SAITS | Solar-10 | 10000 | 10000 | 6h05m |
| SAITS | HAR | 128 | 10000 | 13h35m |
| SAITS | P12 | 48 | 10000 | 10h40m |

Table 17: Training times for imputation task, CSDI.

| Model Name | Dataset | L | Max Epochs | Training Time |
|---|---|---|---|---|
| CSDI | Electricity | 200 | 200 | 2h55m |
| CSDI | Electricity | 2000 | 200 | 6h |
| CSDI | Traffic | 200 | 200 | 3h20m |
| CSDI | Traffic | 2000 | 200 | 7h20m |
| CSDI | Solar-10 | 200 | 200 | 1h30m |
| CSDI | Solar-10 | 10000 | 200 | 12h |
| CSDI | HAR | 128 | 200 | 8h5m |
| CSDI | P12 | 48 | 200 | 16h10m |

Table 18: Training times for forecasting task, TV-INRs.

| Model Name | Dataset | H | Max Epochs | Training Time |
|---|---|---|---|---|
| TV-INR | Electricity | 512 | 2000 | 5h25m |
| TV-INR | Traffic | 512 | 4000 | 11h05m |
| TV-INR | Solar-H | 512 | 2000 | 5h15m |

Table 19: Training times for forecasting task, TimeFlow.

| Model Name | Dataset | H | F | Max Epochs | Training Time |
|---|---|---|---|---|---|
| TimeFlow | Electricity | 512 | 96 | 40000 | 4h25m |
| TimeFlow | Electricity | 512 | 192 | 40000 | 4h30m |
| TimeFlow | Electricity | 512 | 336 | 40000 | 4h40m |
| TimeFlow | Electricity | 512 | 720 | 40000 | 4h30m |
| TimeFlow | Traffic | 512 | 96 | 40000 | 10h10m |
| TimeFlow | Traffic | 512 | 192 | 40000 | 10h15m |
| TimeFlow | Traffic | 512 | 336 | 40000 | 10h20m |
| TimeFlow | Traffic | 512 | 720 | 40000 | 10h15m |
| TimeFlow | Solar-H | 512 | 96 | 40000 | 3h25m |
| TimeFlow | Solar-H | 512 | 192 | 40000 | 2h55m |
| TimeFlow | Solar-H | 512 | 336 | 40000 | 3h05m |
| TimeFlow | Solar-H | 512 | 720 | 40000 | 3h15m |

Table 20: Training times for forecasting task, DeepTime.

| Model Name | Dataset | H | F | Max Epochs | Training Time |
|---|---|---|---|---|---|
| DeepTime | Electricity | 512 | 96 | 50 | 5m |
| DeepTime | Electricity | 512 | 192 | 50 | 5m |
| DeepTime | Electricity | 512 | 336 | 50 | 5m |
| DeepTime | Electricity | 512 | 720 | 50 | 10m |
| DeepTime | Traffic | 512 | 96 | 50 | 10m |
| DeepTime | Traffic | 512 | 192 | 50 | 10m |
| DeepTime | Traffic | 512 | 336 | 50 | 15m |
| DeepTime | Traffic | 512 | 720 | 50 | 15m |
| DeepTime | Solar-H | 512 | 96 | 50 | 5m |
| DeepTime | Solar-H | 512 | 192 | 50 | 5m |
| DeepTime | Solar-H | 512 | 336 | 50 | 5m |
| DeepTime | Solar-H | 512 | 720 | 50 | 5m |

Table 21: **Training Time Efficiency Ratio: TV-INR vs TimeFlow** in hours (h).

| Forecasting Task | | TV-INR | TimeFlow | Ratio (TimeFlow/TV-INR) | |
|---|---|---|---|---|---|
| Dataset | $H$ | Training Time (h) | Cumulative Time (h) | Absolute | Multiplier |
| Electricity | 512 | 5.42 | 18.08 | 12.66 | 3.34× |
| Traffic | 512 | 11.08 | 41.00 | 29.92 | 3.70× |
| Solar | 512 | 5.25 | 12.67 | 7.42 | 2.41× |
| Imputation Task | | TV-INR | TimeFlow | Ratio (TimeFlow/TV-INR) | |
| Dataset | $L$ | Training Time (h) | Cumulative Time (h) | Absolute | Multiplier |
| Electricity | 200 | 8.75 | 19.83 | 11.08 | 2.27× |
| Electricity | 2000 | 12.92 | 16.75 | 3.83 | 1.30× |
| Traffic | 200 | 10.58 | 29.75 | 19.17 | 2.81× |
| Traffic | 2000 | 15.83 | 25.75 | 9.92 | 1.63× |
| Solar | 200 | 10.42 | 19.83 | 9.41 | 1.90× |
| Solar | 10000 | 19.25 | 36.17 | 16.92 | 1.88× |

## A.11 Inference times comparison

We evaluated the computational efficiency of TV-INRs against TimeFlow by measuring inference times on an NVIDIA V100 GPU. Under identical conditions with a batch size of 1, we recorded forward pass execution times in seconds for both models. TimeFlow was configured to use 3 gradient steps during meta-learning, as specified in the original paper (Naour et al., 2024). A key advantage of TV-INRs is that its inference time remains constant, unlike TimeFlow, which exhibits linear scaling with the number of gradient steps performed during meta-learning. This makes TV-INRs particularly attractive for applications requiring consistent and predictable inference latency.

Table 22: Comparison of inference time of TV-INRs and TimeFlow in seconds for imputation task.

| Model | $L$ | $\tau_{\text{Train}}$ | $\tau_{\text{Test}}$ | Electricity Time (s) | Traffic Time (s) | $L$ | Solar-10 Time (s) |
|---|---|---|---|---|---|---|---|
| TimeFlow | 2K | 0.50 | 0.50 | $0.017 \pm 0.001$ | $0.016 \pm 0.001$ | | $0.038 \pm 0.001$ |
| | | 0.30 | 0.30 | $0.016 \pm 0.001$ | $0.016 \pm 0.001$ | 10K | $0.037 \pm 0.001$ |
| | | 0.05 | 0.05 | $0.016 \pm 0.001$ | $0.016 \pm 0.001$ | | $0.037 \pm 0.001$ |
| TimeFlow | 200 | 0.50 | 0.50 | $0.013 \pm 0.001$ | $0.015 \pm 0.001$ | | $0.015 \pm 0.001$ |
| | | 0.30 | 0.30 | $0.012 \pm 0.001$ | $0.015 \pm 0.001$ | 200 | $0.015 \pm 0.001$ |
| | | 0.05 | 0.05 | $0.012 \pm 0.001$ | $0.015 \pm 0.001$ | | $0.015 \pm 0.001$ |
| TV-INRs | 2K | $\sim \mathcal{S}$ | 0.50 | $0.016 \pm 0.001$ | $0.017 \pm 0.001$ | | $0.060 \pm 0.001$ |
| | | | 0.30 | $0.017 \pm 0.001$ | $0.017 \pm 0.001$ | 10K | $0.059 \pm 0.001$ |
| | | | 0.05 | $0.017 \pm 0.001$ | $0.017 \pm 0.001$ | | $0.059 \pm 0.001$ |
| TV-INRs | 200 | $\sim \mathcal{S}$ | 0.50 | $0.014 \pm 0.001$ | $0.013 \pm 0.001$ | | $0.014 \pm 0.001$ |
| | | | 0.30 | $0.014 \pm 0.002$ | $0.013 \pm 0.001$ | 200 | $0.014 \pm 0.001$ |
| | | | 0.05 | $0.014 \pm 0.001$ | $0.013 \pm 0.001$ | | $0.014 \pm 0.001$ |

## A.12 Ablation study on the number of Fourier Frequencies

To empirically quantify the contribution of Fourier Features to the performance of TV-INR, we conduct an ablation study analyzing the model's performance with different numbers of Fourier frequencies ($N_{\text{FF}}$). The experiment is conducted on Electricity dataset for imputation task, and the results are reported, with performance statistics—mean and standard deviation—computed over multiple non-overlapping test windows. The table below presents the Mean Squared Error (MSE) on the imputed values for configurations with $N_{\text{FF}} \in \{256, 128, 32, 0\}$. The results clearly demonstrate that incorporating Fourier Features provides a significant performance benefit, which aligns with findings in the broader literature (Tancik et al., 2020; Dupont et al., 2021). Across all sequence lengths and observation rates, performance degrades substantially as the number of frequencies is reduced, with the best results consistently achieved for $N_{\text{FF}} = 256$.

Table 23: Comparison of inference time of TV-INRs and Timeflow in seconds for forecasting task.

| | | | | Electricity | Traffic | Solar-H |
|---|---|---|---|---|---|---|
| Model | H | $F_{\text{train}}$ | $F_{\text{test}}$ | Time (s) | Time (s) | Time (s) |
| TimeFlow | 512 | 96 | 96 | $0.016 \pm 0.001$ | $0.017 \pm 0.001$ | $0.016 \pm 0.001$ |
| | | 192 | 192 | $0.016 \pm 0.001$ | $0.019 \pm 0.001$ | $0.015 \pm 0.001$ |
| | | 336 | 336 | $0.016 \pm 0.001$ | $0.020 \pm 0.001$ | $0.015 \pm 0.001$ |
| | | 720 | 720 | $0.016 \pm 0.001$ | $0.020 \pm 0.001$ | $0.015 \pm 0.001$ |
| TV-INRs | 512 | $\sim \mathcal{F}$ | 720 | $0.016 \pm 0.001$ | $0.018 \pm 0.001$ | $0.017 \pm 0.002$ |

Table 24: Ablation study on the effect of Fourier Features. We report MSE on the Electricity dataset for different numbers of Fourier Feature frequencies ($N_{\text{FF}}$). The best performing configuration for each row is in bold.

| | | | Number of Fourier Feature Frequencies ($N_{\text{FF}}$) | | | |
|---|---|---|---|---|---|---|
| Model | L | $\tau$ | 256 | 128 | 32 | 0 (None) |
| TV-INRs | 200 | 0.50 | $\mathbf{0.1213 \pm 0.0131}$ | $0.1391 \pm 0.0140$ | $0.1523 \pm 0.0186$ | $0.8099 \pm 0.0522$ |
| | | 0.30 | $\mathbf{0.1359 \pm 0.0265}$ | $0.1756 \pm 0.0211$ | $0.2711 \pm 0.0386$ | $0.8587 \pm 0.0502$ |
| | | 0.05 | $\mathbf{0.3312 \pm 0.0968}$ | $0.4655 \pm 0.1198$ | $0.8643 \pm 0.1206$ | $1.2215 \pm 0.1335$ |
| TV-INRs | 2000 | 0.50 | $\mathbf{0.2555 \pm 0.0280}$ | $0.3563 \pm 0.0236$ | $1.0414 \pm 0.0233$ | $1.0542 \pm 0.0239$ |
| | | 0.30 | $\mathbf{0.2423 \pm 0.0276}$ | $0.3444 \pm 0.0095$ | $1.0341 \pm 0.0503$ | $1.0531 \pm 0.0221$ |
| | | 0.05 | $\mathbf{0.3142 \pm 0.0742}$ | $0.4984 \pm 0.0390$ | $1.0687 \pm 0.0400$ | $1.1004 \pm 0.0278$ |

## A.13 Comparison with standard VAE baseline

To empirically validate the contribution of our Implicit Neural Representation (INR) based decoder, we conduct an ablation study comparing TV-INR against a baseline with a standard decoder, which we term TV-VAE. This baseline is designed to isolate the impact of the INR by replacing the hypernetwork decoder with a conventional MLP. Specifically, the TV-VAE decoder processes a direct concatenation of the learned latent representation $z$ and the time encoding $t$. To ensure a fair comparison, the MLP architecture for the TV-VAE decoder is constructed from the same building blocks as the hypernetwork in TV-INR.

We performed a thorough hyperparameter search for the TV-VAE model, evaluating various MLP depths and multiple configurations of Fourier Features for the time encoding. All other experimental settings, including the AdamW optimizer, followed the protocol used for the main TV-INR experiments as detailed in App. A.6. The results, presented in Tables 25-26. [], show that TV-INR consistently and significantly outperforms all tested variants of TV-VAE on the electricity dataset for sequence lengths $L = 200, 2000$ and across all observation rates ($\tau$). This consistent superiority demonstrates that the INR-based architecture is more effective at modeling the continuous temporal structure of time series signals than a standard decoder that treats time as a concatenated input feature, thereby justifying our architectural choice.

Table 25: Ablation study on the **Electricity dataset (L=200)**. We compare TV-INR with TV-VAE variants using different MLP decoder depths ($D$) and numbers of Fourier Feature frequencies ($N_{FF}$). Best results are in bold.

| Model | $D$ | $N_{FF}$ | $\tau = 0.05$ | | $\tau = 0.3$ | | $\tau = 0.5$ | |
|---|---|---|---|---|---|---|---|---|
| | | | MSE | MAE | MSE | MAE | MSE | MAE |
| TV-VAE | 5 | 256 | $0.98 \pm 0.22$ | $0.78 \pm 0.10$ | $0.44 \pm 0.10$ | $0.48 \pm 0.06$ | $0.34 \pm 0.07$ | $0.41 \pm 0.05$ |
| TV-VAE | 5 | 128 | $1.00 \pm 0.21$ | $0.80 \pm 0.01$ | $0.48 \pm 0.12$ | $0.51 \pm 0.08$ | $0.35 \pm 0.08$ | $0.42 \pm 0.05$ |
| TV-VAE | 5 | 32 | $1.11 \pm 0.39$ | $0.83 \pm 0.16$ | $0.52 \pm 0.16$ | $0.52 \pm 0.09$ | $0.36 \pm 0.10$ | $0.42 \pm 0.06$ |
| TV-VAE | 5 | 0 | $1.24 \pm 0.14$ | $0.83 \pm 0.06$ | $0.52 \pm 0.05$ | $0.50 \pm 0.02$ | $0.43 \pm 0.05$ | $0.45 \pm 0.02$ |
| TV-VAE | 4 | 256 | $0.90 \pm 0.14$ | $0.74 \pm 0.07$ | $0.32 \pm 0.05$ | $0.39 \pm 0.04$ | $0.23 \pm 0.04$ | $0.33 \pm 0.03$ |
| TV-VAE | 4 | 128 | $1.07 \pm 0.14$ | $0.84 \pm 0.06$ | $0.57 \pm 0.08$ | $0.59 \pm 0.05$ | $0.43 \pm 0.07$ | $0.51 \pm 0.04$ |
| TV-VAE | 4 | 32 | $0.65 \pm 0.12$ | $0.61 \pm 0.07$ | $0.25 \pm 0.04$ | $0.34 \pm 0.03$ | $0.20 \pm 0.04$ | $0.30 \pm 0.02$ |
| TV-VAE | 4 | 0 | $1.41 \pm 0.11$ | $0.91 \pm 0.04$ | $0.59 \pm 0.10$ | $0.54 \pm 0.05$ | $0.45 \pm 0.07$ | $0.47 \pm 0.03$ |
| TV-VAE | 3 | 256 | $0.62 \pm 0.16$ | $0.59 \pm 0.08$ | $0.21 \pm 0.04$ | $0.31 \pm 0.03$ | $0.18 \pm 0.03$ | $0.28 \pm 0.02$ |
| TV-VAE | 3 | 128 | $0.50 \pm 0.12$ | $0.43 \pm 0.07$ | $0.19 \pm 0.04$ | $0.28 \pm 0.03$ | $0.17 \pm 0.03$ | $0.27 \pm 0.02$ |
| TV-VAE | 3 | 32 | $0.66 \pm 0.13$ | $0.62 \pm 0.08$ | $0.25 \pm 0.05$ | $0.34 \pm 0.03$ | $0.20 \pm 0.03$ | $0.30 \pm 0.02$ |
| TV-VAE | 3 | 0 | $1.58 \pm 0.27$ | $0.97 \pm 0.08$ | $0.63 \pm 0.09$ | $0.59 \pm 0.04$ | $0.51 \pm 0.06$ | $0.53 \pm 0.03$ |
| TV-VAE | 2 | 256 | $0.88 \pm 0.13$ | $0.78 \pm 0.07$ | $0.45 \pm 0.06$ | $0.53 \pm 0.05$ | $0.34 \pm 0.06$ | $0.44 \pm 0.04$ |
| TV-VAE | 2 | 128 | $0.87 \pm 0.12$ | $0.78 \pm 0.06$ | $0.41 \pm 0.05$ | $0.51 \pm 0.04$ | $0.30 \pm 0.05$ | $0.42 \pm 0.04$ |
| TV-VAE | 2 | 32 | $0.79 \pm 0.20$ | $0.70 \pm 0.10$ | $0.30 \pm 0.05$ | $0.40 \pm 0.04$ | $0.23 \pm 0.04$ | $0.34 \pm 0.03$ |
| TV-VAE | 2 | 0 | $1.59 \pm 0.51$ | $0.97 \pm 0.11$ | $0.84 \pm 0.08$ | $0.71 \pm 0.03$ | $0.76 \pm 0.08$ | $0.67 \pm 0.03$ |
| TV-VAE | 1 | 256 | $0.39 \pm 0.10$ | $0.43 \pm 0.07$ | $0.21 \pm 0.05$ | $0.30 \pm 0.03$ | $0.20 \pm 0.04$ | $0.29 \pm 0.03$ |
| TV-VAE | 1 | 128 | $0.41 \pm 0.06$ | $0.43 \pm 0.08$ | $0.21 \pm 0.05$ | $0.30 \pm 0.03$ | $0.20 \pm 0.04$ | $0.30 \pm 0.03$ |
| TV-VAE | 1 | 32 | $0.39 \pm 0.06$ | $0.44 \pm 0.05$ | $0.23 \pm 0.05$ | $0.32 \pm 0.03$ | $0.22 \pm 0.04$ | $0.31 \pm 0.03$ |
| TV-VAE | 1 | 0 | $1.37 \pm 0.12$ | $0.93 \pm 0.04$ | $1.13 \pm 0.05$ | $0.84 \pm 0.02$ | $1.09 \pm 0.07$ | $0.82 \pm 0.02$ |
| TV-INRs | 3 | 256 | $\mathbf{0.32 \pm 0.06}$ | $\mathbf{0.37 \pm 0.04}$ | $\mathbf{0.14 \pm 0.03}$ | $\mathbf{0.23 \pm 0.02}$ | $\mathbf{0.11 \pm 0.02}$ | $\mathbf{0.21 \pm 0.02}$ |

Table 26: Ablation study on the **Electricity dataset (L=2000)**. We compare TV-INR with TV-VAE variants using different MLP decoder depths ($D$) and numbers of Fourier Feature frequencies ($N_{FF}$). Best results are in bold.

| Model | $D$ | $N_{FF}$ | $\tau = 0.05$ | | $\tau = 0.3$ | | $\tau = 0.5$ | |
|---|---|---|---|---|---|---|---|---|
| | | | MSE | MAE | MSE | MAE | MSE | MAE |
| TV-VAE | 6 | 256 | $0.92 \pm 0.11$ | $0.78 \pm 0.05$ | $0.51 \pm 0.04$ | $0.51 \pm 0.03$ | $0.43 \pm 0.03$ | $0.46 \pm 0.02$ |
| TV-VAE | 6 | 128 | $0.43 \pm 0.06$ | $0.46 \pm 0.03$ | $0.37 \pm 0.04$ | $0.42 \pm 0.03$ | $0.36 \pm 0.04$ | $0.42 \pm 0.03$ |
| TV-VAE | 6 | 32 | $0.94 \pm 0.02$ | $0.74 \pm 0.01$ | $0.89 \pm 0.03$ | $0.71 \pm 0.01$ | $0.89 \pm 0.02$ | $0.71 \pm 0.01$ |
| TV-VAE | 6 | 0 | $1.17 \pm 0.03$ | $0.84 \pm 0.01$ | $1.06 \pm 0.02$ | $0.80 \pm 0.01$ | $1.06 \pm 0.02$ | $0.80 \pm 0.01$ |
| TV-VAE | 5 | 256 | $1.06 \pm 0.23$ | $0.83 \pm 0.11$ | $0.61 \pm 0.07$ | $0.59 \pm 0.05$ | $0.46 \pm 0.04$ | $0.48 \pm 0.02$ |
| TV-VAE | 5 | 128 | $0.44 \pm 0.05$ | $0.46 \pm 0.04$ | $0.38 \pm 0.04$ | $0.43 \pm 0.03$ | $0.37 \pm 0.04$ | $0.42 \pm 0.02$ |
| TV-VAE | 5 | 32 | $0.92 \pm 0.03$ | $0.72 \pm 0.01$ | $0.86 \pm 0.03$ | $0.70 \pm 0.01$ | $0.86 \pm 0.03$ | $0.70 \pm 0.01$ |
| TV-VAE | 5 | 0 | $1.16 \pm 0.03$ | $0.84 \pm 0.01$ | $1.05 \pm 0.02$ | $0.80 \pm 0.01$ | $1.05 \pm 0.02$ | $0.80 \pm 0.01$ |
| TV-VAE | 4 | 256 | $0.33 \pm 0.02$ | $0.39 \pm 0.02$ | $0.28 \pm 0.02$ | $0.36 \pm 0.01$ | $0.26 \pm 0.02$ | $0.35 \pm 0.01$ |
| TV-VAE | 4 | 128 | $0.35 \pm 0.03$ | $0.41 \pm 0.02$ | $0.32 \pm 0.02$ | $0.39 \pm 0.01$ | $0.32 \pm 0.02$ | $0.39 \pm 0.01$ |
| TV-VAE | 4 | 32 | $0.75 \pm 0.02$ | $0.67 \pm 0.02$ | $0.72 \pm 0.02$ | $0.65 \pm 0.02$ | $0.72 \pm 0.03$ | $0.65 \pm 0.02$ |
| TV-VAE | 4 | 0 | $1.10 \pm 0.01$ | $0.83 \pm 0.01$ | $1.04 \pm 0.02$ | $0.80 \pm 0.01$ | $1.05 \pm 0.02$ | $0.80 \pm 0.01$ |
| TV-VAE | 3 | 256 | $0.37 \pm 0.02$ | $0.43 \pm 0.02$ | $0.33 \pm 0.02$ | $0.40 \pm 0.02$ | $0.32 \pm 0.03$ | $0.40 \pm 0.02$ |
| TV-VAE | 3 | 128 | $0.43 \pm 0.05$ | $0.48 \pm 0.04$ | $0.40 \pm 0.04$ | $0.46 \pm 0.03$ | $0.40 \pm 0.04$ | $0.46 \pm 0.03$ |
| TV-VAE | 3 | 32 | $0.98 \pm 0.01$ | $0.80 \pm 0.01$ | $0.91 \pm 0.01$ | $0.77 \pm 0.01$ | $0.91 \pm 0.01$ | $0.76 \pm 0.01$ |
| TV-VAE | 3 | 0 | $1.09 \pm 0.01$ | $0.82 \pm 0.01$ | $1.04 \pm 0.03$ | $0.80 \pm 0.01$ | $1.05 \pm 0.02$ | $0.80 \pm 0.01$ |
| TV-VAE | 2 | 256 | $0.34 \pm 0.03$ | $0.41 \pm 0.02$ | $0.31 \pm 0.02$ | $0.38 \pm 0.01$ | $0.30 \pm 0.02$ | $0.38 \pm 0.01$ |
| TV-VAE | 2 | 128 | $0.56 \pm 0.08$ | $0.58 \pm 0.05$ | $0.53 \pm 0.07$ | $0.56 \pm 0.05$ | $0.53 \pm 0.08$ | $0.56 \pm 0.05$ |
| TV-VAE | 2 | 32 | $1.05 \pm 0.01$ | $0.82 \pm 0.01$ | $1.01 \pm 0.03$ | $0.79 \pm 0.01$ | $1.01 \pm 0.02$ | $0.79 \pm 0.01$ |
| TV-VAE | 2 | 0 | $1.08 \pm 0.01$ | $0.81 \pm 0.01$ | $1.06 \pm 0.03$ | $0.80 \pm 0.01$ | $1.06 \pm 0.02$ | $0.80 \pm 0.01$ |
| TV-INRs | 4 | 256 | $\mathbf{0.29 \pm 0.02}$ | $\mathbf{0.36 \pm 0.02}$ | $\mathbf{0.25 \pm 0.02}$ | $\mathbf{0.33 \pm 0.01}$ | $\mathbf{0.25 \pm 0.02}$ | $\mathbf{0.33 \pm 0.01}$ |

## B  Appendix B

### B.1  Uncertainty evaluation

Table 27: Uncertainty and calibration of TV-INRs evaluation on the Electricity and HAR imputation tasks. We report negative log-likelihood (NLL) and empirical coverage of the nominal 90% prediction interval, computed in normalized space.

| Dataset | $L$ | $\tau$ | Coverage@90 ↑ | NLL ↓ |
|---|---|---|---|---|
| Electricity | 200 | 0.95 | 0.70 | 0.19 |
| | | 0.50 | 0.58 | 1.18 |
| | | 0.30 | 0.63 | 1.37 |
| | | 0.05 | 0.45 | 3.35 |
| | 2000 | 0.95 | 0.73 | 0.78 |
| | | 0.50 | 0.61 | 1.75 |
| | | 0.30 | 0.59 | 1.82 |
| | | 0.05 | 0.58 | 1.97 |
| HAR | 128 | 0.95 | 0.62 | 2.14 |
| | | 0.50 | 0.60 | 2.41 |
| | | 0.30 | 0.58 | 2.43 |
| | | 0.05 | 0.51 | 3.25 |

Table 28: Uncertainty and calibration evaluation on the Electricity forecasting task. We report negative log-likelihood and empirical coverage of the nominal 90% prediction interval, computed in normalized space.

| Dataset | Forecast Horizon | Coverage@90 ↑ | NLL ↓ |
|---|---|---|---|
| Electricity | 96 | 0.66 | 0.66 |
| | 192 | 0.60 | 1.03 |
| | 336 | 0.59 | 1.10 |
| | 720 | 0.58 | 1.17 |

We evaluate predictive uncertainty using negative log-likelihood (NLL) and empirical coverage of the nominal 90% prediction interval. All metrics are computed in normalized space. Table 27 summarizes uncertainty and calibration results for univariate imputation on the Electricity dataset at two sequence lengths ($L = 200$ and $L = 2000$) and for multivariate imputation on the HAR dataset, evaluated under different observation rates. As missingness increases, predictive NLL rises while empirical coverage decreases on average, reflecting growing uncertainty under data scarcity. For short sequences (L = 200), coverage degrades substantially under extreme sparsity, accompanied by a sharp increase in NLL. The increase in coverage observed at $\tau = 0.3$ may indicate a calibration effect: the model appears to widen its predictive intervals, leading to higher empirical coverage at the cost of increased NLL. This behavior suggests that the model recognizes its reduced confidence under partial observability and compensates by expressing greater uncertainty, achieving improved coverage but reduced sharpness. In contrast, longer sequences ($L = 2000$) exhibit more stable behavior across missingness levels. While coverage does not uniformly improve at moderate observation rates, the degradation in both empirical coverage and NLL is noticeably more gradual compared to shorter sequences. This effect is particularly evident under extreme sparsity ($\tau = 0.05$), where $L = 200$ suffers a sharp collapse in coverage and a large NLL increase, whereas $L = 2000$ remains comparatively well-performed. These results suggest that additional temporal context primarily improves robustness rather than uniformly enhancing calibration. Similar behavior is observed on the HAR dataset. As the observation rate decreases, empirical coverage declines smoothly while NLL increases accordingly, mirroring the Electricity trends. The consistency of these degradation patterns across sequence lengths and datasets indicates that TV-INRs' uncertainty estimates respond coherently to increasing task difficulty. Although absolute coverage remains imperfect, the relative stability across regimes points to a latent representation that degrades gracefully under partial observability. Table 28 reports uncertainty metrics for Electricity forecasting across different prediction horizons. As the

forecast horizon increases, predictive NLL rises while empirical coverage of the nominal 90% prediction interval decreases, reflecting the growing uncertainty associated with long-range forecasts.

## B.2 Detailed analysis of forecasting errors

To better understand the source of large error spikes at long forecast horizons for Electricity (Table 2), we analyze the distribution of absolute prediction errors across different horizons as shown in Figure 4. For each forecast horizon, we aggregate absolute errors over all test windows and visualize their empirical distributions using histograms. We annotate each distribution with the median absolute error, mean absolute error (MAE), and the 95th percentile of absolute error to distinguish typical behavior from extreme deviations.

The resulting distributions indicate that median error remains relatively stable across horizons, while MAE increases only moderately. In contrast, the upper tail of the error distribution becomes progressively heavier as the forecast horizon increases, with the effect being most pronounced at the longest horizon (F = 720). It exhibits a small number of extreme outliers with very large absolute errors. These exceptional but severe failures dominate squared-error metrics, explaining the sharp increase in MSE at long horizons despite relatively stable typical performance.

## B.3 Forecasting with partially observed history

We evaluate also forecasting under *partially observed history*: unlike the standard setting where the full look-back window is available, a fraction of the history time steps is missing at prediction time. We use the hourly Electricity dataset with a look-back of $T_h$=512 steps and evaluate forecasts at horizons $L \in \{96, 192, 336, 720\}$. Missingness is applied *only* to the history window; the forecast horizon is always the prediction target and is never observed. At evaluation we sweep the observed-history ratio $\rho \in \{0.5, 0.2, 0.1\}$, i.e. only 50%, 20%, or 10% of the history steps are retained (chosen with a fixed, model-independent random mask so all methods see the same observed sub-sampling). We show the results in Table 29.

Crucially, TV-INRs is a *single* model: it is trained once with the forecast length sampled from a distribution ($F \sim \mathcal{F}$), and the *same* trained model is then evaluated across all horizons $L$ *and* all missingness ratios. In contrast, the baselines are trained *per case*: a separate DeepTime / TimeFlow model is fit for each forecast horizon, on the full-history regime. TV-INRs thus solves a strictly harder problem—one model must cover every horizon and every level of history sparsity—yet remains competitive with, and under heavy missingness superior to, the horizon-specialised baselines.

As the observed-history ratio decreases, every method degrades, but not equally. DeepTime, which zero-fills the unobserved history, deteriorates most sharply. TimeFlow, an implicit model fit to the observed points, degrades gracefully and remains a strong baseline at mild-to-moderate sparsity. TV-INRs matches this robustness while additionally benefiting from its missing-history training curriculum: under the most severe setting ($\tau_{\text{hist}}$=0.1, only 10% of the look-back observed) it attains the best forecast accuracy across horizons. These results indicate that representing each series as a latent-conditioned implicit function, trained directly under partial observation, yields graceful degradation to sparse look-back windows—without the need to train a dedicated model per horizon.

## B.4 Likelihood sensitivity on forecasting task

We perform a sensitivity analysis comparing the Laplace likelihood used in our main experiments with a Gaussian likelihood to examine the effect of the observation model on forecasting behavior (Table 30). Overall performance trends are similar across likelihood choices, with central error metrics such as MAE and median absolute error remaining comparable. At longer forecast horizons, the Gaussian likelihood shows slightly increased sensitivity to large prediction errors, while the Laplace likelihood exhibits marginally reduced influence from extreme values. These differences are most apparent in squared-error metrics and suggest that the observed long-horizon behavior is primarily driven by tail effects, with likelihood choice having a secondary impact.

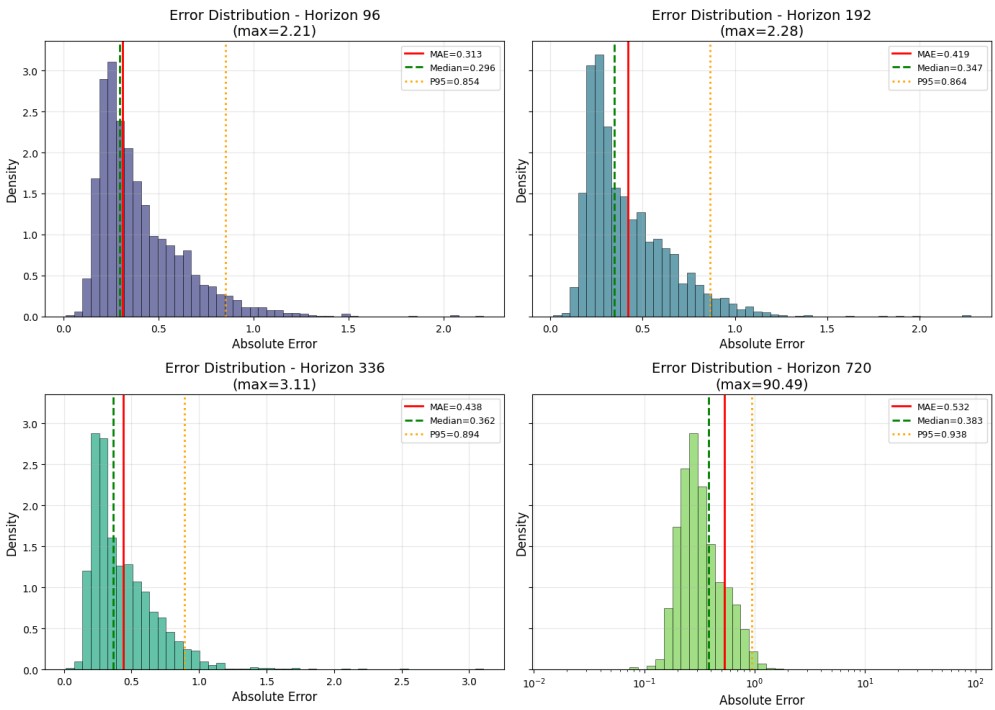

Figure 4: The histograms show the empirical distribution of absolute forecasting errors produced by TV-INRs across different forecast horizons, aggregated over all test windows of the Electricity dataset for a single experimental seed. Vertical lines denote the median absolute error, mean absolute error (MAE), and the 95th percentile of absolute error.

Table 29: **Forecasting with missing history** (Electricity, $H$=512). $\tau_{\text{hist}}$ is the observed-history ratio ($\tau_{\text{hist}}$=1.0: full look-back; $\tau_{\text{hist}}$=0.1: only 10% observed). Baselines are trained on full history; TV-INRs is trained with a missing-history curriculum. All methods share the same evaluation masks. MSE/MAE on the forecast region; bold: best, underline: second best (lower is better).

| Model | H | F | $\tau_{\text{hist}} = 1.0$ | | $\tau_{\text{hist}} = 0.5$ | | $\tau_{\text{hist}} = 0.3$ | | $\tau_{\text{hist}} = 0.1$ | |
| | | | MSE | MAE | MSE | MAE | MSE | MAE | MSE | MAE |
|---|---|---|---|---|---|---|---|---|---|---|
| DeepTime | 512 | 96 | $0.436 \pm 0.020$ | $0.503 \pm 0.016$ | $0.621 \pm 0.098$ | $0.628 \pm 0.061$ | $0.934 \pm 0.171$ | $0.812 \pm 0.102$ | $1.587 \pm 0.312$ | $1.124 \pm 0.168$ |
| | | 192 | $0.551 \pm 0.157$ | $0.525 \pm 0.055$ | $0.763 \pm 0.201$ | $0.664 \pm 0.088$ | $1.124 \pm 0.263$ | $0.857 \pm 0.121$ | $1.832 \pm 0.388$ | $1.156 \pm 0.184$ |
| | | 336 | $\underline{0.793 \pm 0.046}$ | $0.689 \pm 0.037$ | $\underline{1.087 \pm 0.132}$ | $0.842 \pm 0.076$ | $\underline{1.598 \pm 0.241}$ | $1.078 \pm 0.128$ | $\underline{2.487 \pm 0.403}$ | $1.432 \pm 0.192$ |
| | | 720 | $10.178 \pm 0.218$ | $0.970 \pm 0.178$ | $12.764 \pm 0.301$ | $1.187 \pm 0.212$ | $16.842 \pm 0.402$ | $1.498 \pm 0.256$ | $22.413 \pm 0.548$ | $1.934 \pm 0.321$ |
| TimeFlow | 512 | 96 | $\underline{0.425 \pm 0.057}$ | $\underline{0.318 \pm 0.050}$ | $\underline{0.497 \pm 0.096}$ | $\underline{0.352 \pm 0.064}$ | $\underline{0.601 \pm 0.142}$ | $\underline{0.419 \pm 0.088}$ | $\underline{0.812 \pm 0.233}$ | $\underline{0.547 \pm 0.121}$ |
| | | 192 | $\underline{0.498 \pm 0.078}$ | $\mathbf{0.362 \pm 0.060}$ | $\underline{0.579 \pm 0.128}$ | $\mathbf{0.402 \pm 0.074}$ | $\underline{0.704 \pm 0.176}$ | $\mathbf{0.478 \pm 0.101}$ | $\underline{0.963 \pm 0.284}$ | $\underline{0.628 \pm 0.143}$ |
| | | 336 | $1.347 \pm 0.210$ | $\mathbf{0.389 \pm 0.065}$ | $1.541 \pm 0.248$ | $\mathbf{0.451 \pm 0.083}$ | $1.872 \pm 0.331$ | $\mathbf{0.538 \pm 0.108}$ | $2.518 \pm 0.442$ | $\underline{0.771 \pm 0.158}$ |
| | | 720 | $\mathbf{9.422 \pm 0.217}$ | $\mathbf{0.525 \pm 0.150}$ | $10.214 \pm 0.263$ | $\mathbf{0.581 \pm 0.168}$ | $12.103 \pm 0.352$ | $\mathbf{0.682 \pm 0.201}$ | $\underline{16.488 \pm 0.521}$ | $\underline{0.846 \pm 0.258}$ |
| TV-INRs | 512 | 96 | $\mathbf{0.336 \pm 0.068}$ | $\mathbf{0.296 \pm 0.040}$ | $\mathbf{0.382 \pm 0.075}$ | $\mathbf{0.321 \pm 0.045}$ | $\mathbf{0.454 \pm 0.092}$ | $\mathbf{0.355 \pm 0.052}$ | $\mathbf{0.623 \pm 0.140}$ | $\mathbf{0.454 \pm 0.078}$ |
| | | 192 | $\mathbf{0.446 \pm 0.107}$ | $\underline{0.415 \pm 0.036}$ | $\mathbf{0.510 \pm 0.118}$ | $\underline{0.442 \pm 0.042}$ | $\mathbf{0.622 \pm 0.150}$ | $\underline{0.493 \pm 0.055}$ | $\mathbf{0.858 \pm 0.205}$ | $\mathbf{0.621 \pm 0.096}$ |
| | | 336 | $\mathbf{0.544 \pm 0.216}$ | $\underline{0.442 \pm 0.040}$ | $\mathbf{0.623 \pm 0.230}$ | $\underline{0.478 \pm 0.048}$ | $\mathbf{0.782 \pm 0.270}$ | $\underline{0.547 \pm 0.062}$ | $\mathbf{1.251 \pm 0.360}$ | $\mathbf{0.753 \pm 0.105}$ |
| | | 720 | $\underline{9.515 \pm 0.218}$ | $\underline{0.535 \pm 0.162}$ | $\underline{10.500 \pm 0.240}$ | $\underline{0.590 \pm 0.170}$ | $\underline{12.801 \pm 0.300}$ | $\underline{0.687 \pm 0.190}$ | $\mathbf{16.102 \pm 0.420}$ | $\mathbf{0.822 \pm 0.230}$ |

Table 30: **Likelihood sensitivity on Electricity forecasting.** We compare Laplace (default) and Gaussian likelihoods for TV-INRs across forecast horizons. MSE and MAE are reported on the forecasting task.

| Model | Likelihood | $H$ | $F_{\text{test}}$ | MSE | MAE |
|---|---|---|---|---|---|
| TV-INRs | Laplace | 512 | 96 | $0.336 \pm 0.068$ | $0.296 \pm 0.040$ |
| | | | 192 | $0.446 \pm 0.107$ | $0.415 \pm 0.036$ |
| | | | 336 | $0.544 \pm 0.216$ | $0.442 \pm 0.040$ |
| | | | 720 | $9.515 \pm 0.218$ | $0.535 \pm 0.162$ |
| TV-INRs | Gaussian | 512 | 96 | $0.431 \pm 0.064$ | $0.444 \pm 0.034$ |
| | | | 192 | $0.478 \pm 0.096$ | $0.455 \pm 0.034$ |
| | | | 336 | $0.570 \pm 0.206$ | $0.475 \pm 0.036$ |
| | | | 720 | $9.543 \pm 0.217$ | $0.564 \pm 0.155$ |

## B.5 Statistical analysis of the results

Best models are determined by mean performance of the corresponding experiment. Significance levels are denotes as: *** $p < 0.001$, ** $p < 0.01$, * $p < 0.05$, ns: not significant.

### B.5.1 Univariate Imputation

We report the statistical analysis of imputation results in Table 31.

**Electricity.** In high-data regimes ($L = 2000$, $\tau \geq 0.30$), TimeFlow significantly outperforms all baselines, reflecting the strength of per-instance optimization when abundant observations are available. However, under extreme sparsity ($\tau = 0.05$) and across all short-sequence settings ($L = 200$), TV-INRs achieves statistically significant improvements over all competitors, including TimeFlow.

**Traffic.** In high-data regimes ($L = 2000$, $\tau \geq 0.30$), SAITS significantly outperforms competing models, reflecting the effectiveness of attention-based architectures under dense observations. Under severe sparsity ($\tau = 0.05$) and across all short-sequence settings ($L = 200$), TV-INRs consistently achieves statistically significant improvements over SAITS and CSDI, and in most cases over TimeFlow.

**Solar-10.** For very long sequences ($L = 10000$), TimeFlow achieves significantly lower MSE in most regimes, consistent with the benefits of per-instance optimization when extensive temporal structure is available. However, these gains do not consistently extend to MAE. When sequence length is reduced ($L = 200$), TV-INRs consistently outperforms all baselines across all observation ratios with statistically significant margins.

Table 31: **Statistical Results — Imputation Task — Electricity, Traffic, and Solar-10 Datasets.**

**Electricity Dataset**

| L | τ | Metric | Best | vs. | Sig. |
|---|---|---|---|---|---|
| 2000 | 0.50 | MSE | TimeFlow | SAITS | *** |
| | | | TimeFlow | CSDI | *** |
| | | | TimeFlow | TV-INRs | *** |
| | | MAE | TimeFlow | SAITS | *** |
| | | | TimeFlow | CSDI | *** |
| | | | TimeFlow | TV-INRs | *** |
| | 0.30 | MSE | TimeFlow | SAITS | *** |
| | | | TimeFlow | CSDI | *** |
| | | | TimeFlow | TV-INRs | *** |
| | | MAE | TimeFlow | SAITS | *** |
| | | | TimeFlow | CSDI | *** |
| | | | TimeFlow | TV-INRs | *** |
| | 0.05 | MSE | TV-INRs | SAITS | *** |
| | | | TV-INRs | CSDI | *** |
| | | | TV-INRs | TimeFlow | ** |
| | | MAE | TV-INRs | SAITS | *** |
| | | | TV-INRs | CSDI | *** |
| | | | TV-INRs | TimeFlow | *** |
| 200 | 0.50 | MSE | TV-INRs | SAITS | *** |
| | | | TV-INRs | CSDI | *** |
| | | | TV-INRs | TimeFlow | *** |
| | | MAE | TV-INRs | SAITS | *** |
| | | | TV-INRs | CSDI | *** |
| | | | TV-INRs | TimeFlow | *** |
| | 0.30 | MSE | TV-INRs | SAITS | *** |
| | | | TV-INRs | CSDI | *** |
| | | | TV-INRs | TimeFlow | *** |
| | | MAE | TV-INRs | SAITS | *** |
| | | | TV-INRs | CSDI | *** |
| | | | TV-INRs | TimeFlow | *** |
| | 0.05 | MSE | TV-INRs | SAITS | *** |
| | | | TV-INRs | CSDI | *** |
| | | | TV-INRs | TimeFlow | *** |
| | | MAE | TV-INRs | SAITS | *** |
| | | | TV-INRs | CSDI | *** |
| | | | TV-INRs | TimeFlow | *** |

**Traffic Dataset**

| L | τ | Metric | Best | vs. | Sig. |
|---|---|---|---|---|---|
| 2000 | 0.50 | MSE | SAITS | CSDI | *** |
| | | | SAITS | TimeFlow | *** |
| | | | SAITS | TV-INRs | *** |
| | | MAE | SAITS | CSDI | *** |
| | | | SAITS | TimeFlow | *** |
| | | | SAITS | TV-INRs | *** |
| | 0.30 | MSE | SAITS | CSDI | *** |
| | | | SAITS | TimeFlow | * |
| | | | SAITS | TV-INRs | *** |
| | | MAE | SAITS | CSDI | *** |
| | | | SAITS | TimeFlow | *** |
| | | | SAITS | TV-INRs | *** |
| | 0.05 | MSE | TV-INRs | SAITS | *** |
| | | | TV-INRs | CSDI | *** |
| | | | TV-INRs | TimeFlow | ns |
| | | MAE | TV-INRs | SAITS | *** |
| | | | TV-INRs | CSDI | ** |
| | | | TV-INRs | TimeFlow | *** |
| 200 | 0.50 | MSE | TV-INRs | SAITS | *** |
| | | | TV-INRs | CSDI | *** |
| | | | TV-INRs | TimeFlow | *** |
| | | MAE | TV-INRs | SAITS | *** |
| | | | TV-INRs | CSDI | *** |
| | | | TV-INRs | TimeFlow | *** |
| | 0.30 | MSE | TV-INRs | SAITS | *** |
| | | | TV-INRs | CSDI | *** |
| | | | TV-INRs | TimeFlow | *** |
| | | MAE | TV-INRs | SAITS | *** |
| | | | TV-INRs | CSDI | *** |
| | | | TV-INRs | TimeFlow | *** |
| | 0.05 | MSE | TV-INRs | SAITS | *** |
| | | | TV-INRs | CSDI | *** |
| | | | TV-INRs | TimeFlow | *** |
| | | MAE | TV-INRs | SAITS | *** |
| | | | TV-INRs | CSDI | *** |
| | | | TV-INRs | TimeFlow | *** |

**Solar-10 Dataset**

| L | τ | Metric | Best | vs. | Sig. |
|---|---|---|---|---|---|
| 10000 | 0.50 | MSE | TimeFlow | SAITS | *** |
| | | | TimeFlow | CSDI | * |
| | | | TimeFlow | TV-INRs | *** |
| | | MAE | TimeFlow | SAITS | ns |
| | | | TimeFlow | CSDI | ns |
| | | | TimeFlow | TV-INRs | ns |
| | 0.30 | MSE | TimeFlow | SAITS | ** |
| | | | TimeFlow | CSDI | *** |
| | | | TimeFlow | TV-INRs | * |
| | | MAE | TV-INRs | SAITS | ns |
| | | | TV-INRs | CSDI | ns |
| | | | TV-INRs | TimeFlow | ns |
| | 0.05 | MSE | TimeFlow | SAITS | *** |
| | | | TimeFlow | CSDI | *** |
| | | | TimeFlow | TV-INRs | ns |
| | | MAE | TimeFlow | SAITS | ns |
| | | | TimeFlow | CSDI | ns |
| | | | TimeFlow | TV-INRs | ns |
| 200 | 0.50 | MSE | TV-INRs | SAITS | *** |
| | | | TV-INRs | CSDI | *** |
| | | | TV-INRs | TimeFlow | *** |
| | | MAE | TV-INRs | SAITS | *** |
| | | | TV-INRs | CSDI | *** |
| | | | TV-INRs | TimeFlow | *** |
| | 0.30 | MSE | TV-INRs | SAITS | *** |
| | | | TV-INRs | CSDI | *** |
| | | | TV-INRs | TimeFlow | *** |
| | | MAE | TV-INRs | SAITS | *** |
| | | | TV-INRs | CSDI | *** |
| | | | TV-INRs | TimeFlow | *** |
| | 0.05 | MSE | TV-INRs | SAITS | *** |
| | | | TV-INRs | CSDI | *** |
| | | | TV-INRs | TimeFlow | *** |
| | | MAE | TV-INRs | SAITS | *** |
| | | | TV-INRs | CSDI | *** |
| | | | TV-INRs | TimeFlow | *** |

### B.5.2 Univariate Forecasting

We report the statistical analysis of forecasting results in Table 32.

**Electricity.** For short horizons ($F = 96$), TV-INRs significantly outperforms both DeepTime and TimeFlow in MSE and MAE. At intermediate horizons ($F = 192, 336$), performance becomes mixed: TV-INRs achieves significantly lower MSE, while TimeFlow performs better in MAE. For long-range forecasting ($F = 720$), TimeFlow attains the lowest errors, though differences with TV-INRs are not statistically significant in several cases.

**Traffic.** TimeFlow dominates at short horizons ($F = 96$) and remains competitive at $F = 192$. At medium horizons ($F = 336$), TV-INRs achieves significantly lower MAE and competitive MSE. For long horizons ($F = 720$), TimeFlow attains lower MSE, while TV-INRs maintains competitive MAE. Overall, results show complementary strengths, with TV-INRs particularly competitive in MAE at medium forecasting lengths.

**Solar-H.** TV-INRs achieves the strongest performance at short horizons ($F = 96$), significantly outperforming DeepTime and matching TimeFlow. At longer horizons, TimeFlow consistently achieves lower errors, particularly beyond $F = 336$. This pattern mirrors the imputation findings: amortized inference excels at shorter prediction ranges, while per-instance adaptation becomes increasingly beneficial for extended horizons.

### B.5.3 Multivariate Imputation

We report the statistical analysis of multivariate imputation results in Table 33.

**HAR.** Across all observation ratios, C-TV-INRs significantly outperforms SAITS and CSDI in both MSE and MAE. The conditional formulation consistently matches or slightly improves over the unconditional TV-INRs, with statistically significant gains emerging under extreme sparsity ($\tau = 0.05$). These results confirm that incorporating covariates enhances individualization in dense multivariate settings.

Table 32: **Statistical Results — Forecasting Task (H=512).**

| | Electricity | | | | | Traffic | | | | | Solar-H | | | |
|---|---|---|---|---|---|---|---|---|---|---|---|---|---|---|
| $F$ | Met. | Best | vs. | Sig. | $F$ | Met. | Best | vs. | Sig. | $F$ | Met. | Best | vs. | Sig. |
| 96 | MSE | TV-INRs | DeepTime | *** | 96 | MSE | TimeFlow | DeepTime | *** | 96 | MSE | TV-INRs | DeepTime | ** |
| | | TV-INRs | TimeFlow | *** | | | TimeFlow | TV-INRs | * | | | TV-INRs | TimeFlow | ns |
| | MAE | TV-INRs | DeepTime | *** | | MAE | TimeFlow | DeepTime | *** | | MAE | TV-INRs | DeepTime | *** |
| | | TV-INRs | TimeFlow | ns | | | TimeFlow | TV-INRs | ns | | | TV-INRs | TimeFlow | ns |
| 192 | MSE | TV-INRs | DeepTime | * | 192 | MSE | TimeFlow | DeepTime | ** | 192 | MSE | DeepTime | TimeFlow | ns |
| | | TV-INRs | TimeFlow | ns | | | TimeFlow | TV-INRs | ns | | | DeepTime | TV-INRs | ns |
| | MAE | TimeFlow | DeepTime | *** | | MAE | TV-INRs | DeepTime | *** | | MAE | TimeFlow | DeepTime | *** |
| | | TimeFlow | TV-INRs | ** | | | TV-INRs | TimeFlow | ns | | | TimeFlow | TV-INRs | ns |
| 336 | MSE | TV-INRs | DeepTime | *** | 336 | MSE | TV-INRs | DeepTime | * | 336 | MSE | TimeFlow | DeepTime | *** |
| | | TV-INRs | TimeFlow | *** | | | TV-INRs | TimeFlow | ns | | | TimeFlow | TV-INRs | ns |
| | MAE | TimeFlow | DeepTime | *** | | MAE | TV-INRs | DeepTime | *** | | MAE | TimeFlow | DeepTime | *** |
| | | TimeFlow | TV-INRs | ** | | | TV-INRs | TimeFlow | * | | | TimeFlow | TV-INRs | *** |
| 720 | MSE | TimeFlow | DeepTime | *** | 720 | MSE | TimeFlow | DeepTime | *** | 720 | MSE | TimeFlow | DeepTime | *** |
| | | TimeFlow | TV-INRs | ns | | | TimeFlow | TV-INRs | ns | | | TimeFlow | TV-INRs | * |
| | MAE | TimeFlow | DeepTime | *** | | MAE | TV-INRs | DeepTime | *** | | MAE | TimeFlow | DeepTime | *** |
| | | TimeFlow | TV-INRs | ns | | | TV-INRs | TimeFlow | ns | | | TimeFlow | TV-INRs | ** |

Table 33: **Statistical Results — Multivariate Imputation.**

| | HAR (L=128) | | | | | P12 (L=48) | | | |
|---|---|---|---|---|---|---|---|---|---|
| $\tau$ | Metric | Best | vs. | Sig. | $\tau$ | Metric | Best | vs. | Sig. |
| 0.50 | MSE | C-TV-INRs | SAITS | *** | 0.50 | MSE | TV-INRs | SAITS | *** |
| | | C-TV-INRs | CSDI | *** | | | TV-INRs | CSDI | ns |
| | | C-TV-INRs | TV-INRs | ns | | | TV-INRs | C-TV-INRs | ns |
| | MAE | C-TV-INRs | SAITS | *** | | MAE | TV-INRs | SAITS | *** |
| | | C-TV-INRs | CSDI | *** | | | TV-INRs | CSDI | ** |
| | | C-TV-INRs | TV-INRs | ns | | | TV-INRs | C-TV-INRs | ns |
| 0.30 | MSE | C-TV-INRs | SAITS | *** | 0.30 | MSE | C-TV-INRs | SAITS | *** |
| | | C-TV-INRs | CSDI | *** | | | C-TV-INRs | CSDI | * |
| | | C-TV-INRs | TV-INRs | ns | | | C-TV-INRs | TV-INRs | ns |
| | MAE | C-TV-INRs | SAITS | *** | | MAE | C-TV-INRs | SAITS | *** |
| | | C-TV-INRs | CSDI | *** | | | C-TV-INRs | CSDI | *** |
| | | C-TV-INRs | TV-INRs | ns | | | C-TV-INRs | TV-INRs | ns |
| 0.05 | MSE | C-TV-INRs | SAITS | *** | 0.10 | MSE | C-TV-INRs | SAITS | ns |
| | | C-TV-INRs | CSDI | *** | | | C-TV-INRs | CSDI | *** |
| | | C-TV-INRs | TV-INRs | * | | | C-TV-INRs | TV-INRs | ns |
| | MAE | C-TV-INRs | SAITS | *** | | MAE | C-TV-INRs | SAITS | ns |
| | | C-TV-INRs | CSDI | *** | | | C-TV-INRs | CSDI | *** |
| | | C-TV-INRs | TV-INRs | *** | | | C-TV-INRs | TV-INRs | ns |

**P12.** In moderately observed settings ($\tau = 0.50$), TV-INRs achieves the lowest errors, significantly outperforming SAITS and matching CSDI and C-TV-INRs. As sparsity increases ($\tau = 0.30$ and $0.10$), the conditional formulation (C-TV-INRs) consistently achieves the best performance, particularly against CSDI. These findings demonstrate that incorporating static covariates becomes increasingly beneficial in highly sparse clinical time series.

## B.6    Visuals from experiments

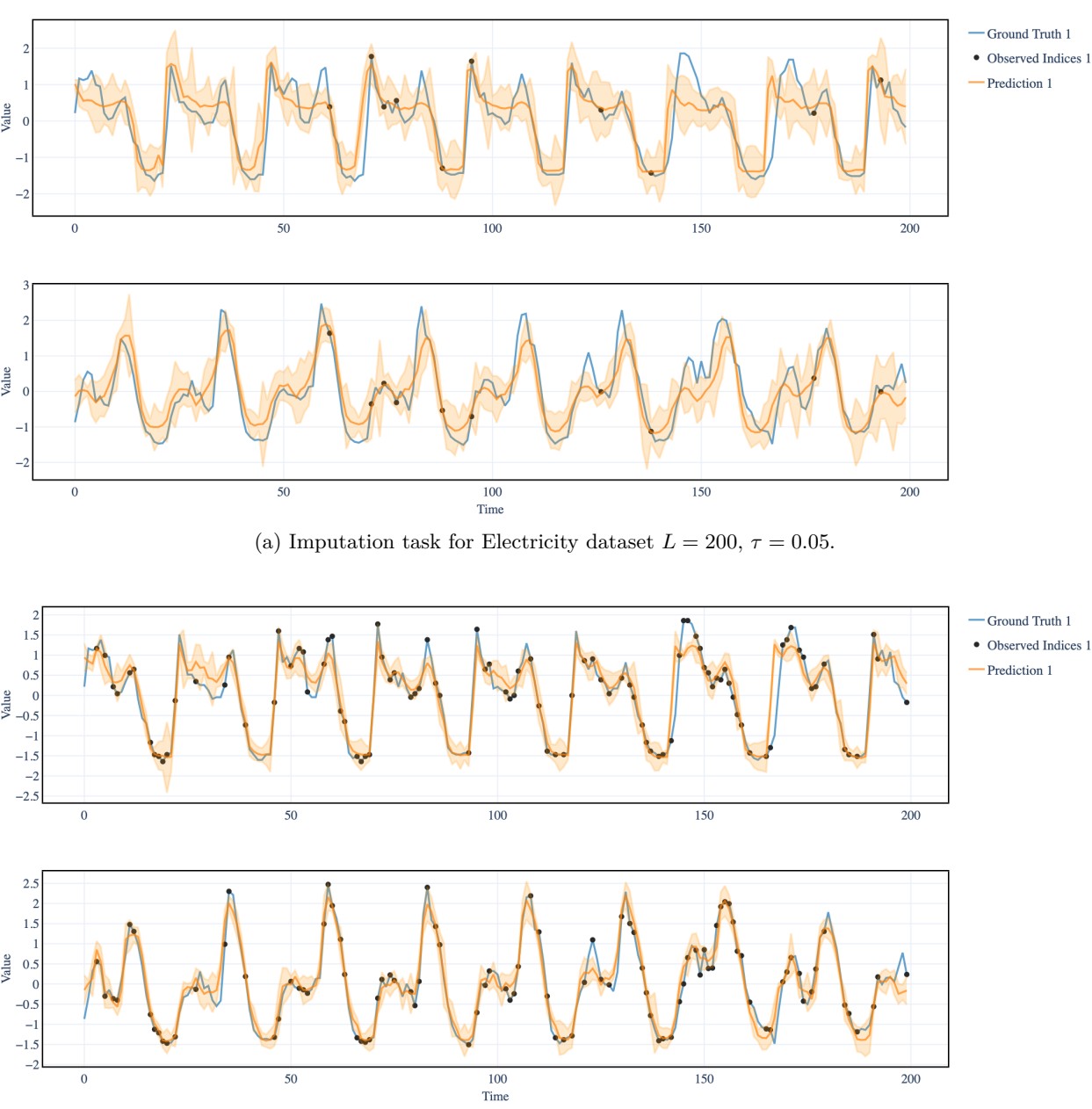

(a) Imputation task for Electricity dataset $L = 200$, $\tau = 0.05$.

(b) Imputation task for Electricity dataset $L = 200$, $\tau = 0.5$.

Figure 5: TV-INRs imputation predictions for Electricity dataset ($L = 200$). Solid lines denote the posterior mean and shaded regions correspond to the 5th–95th percentile intervals.

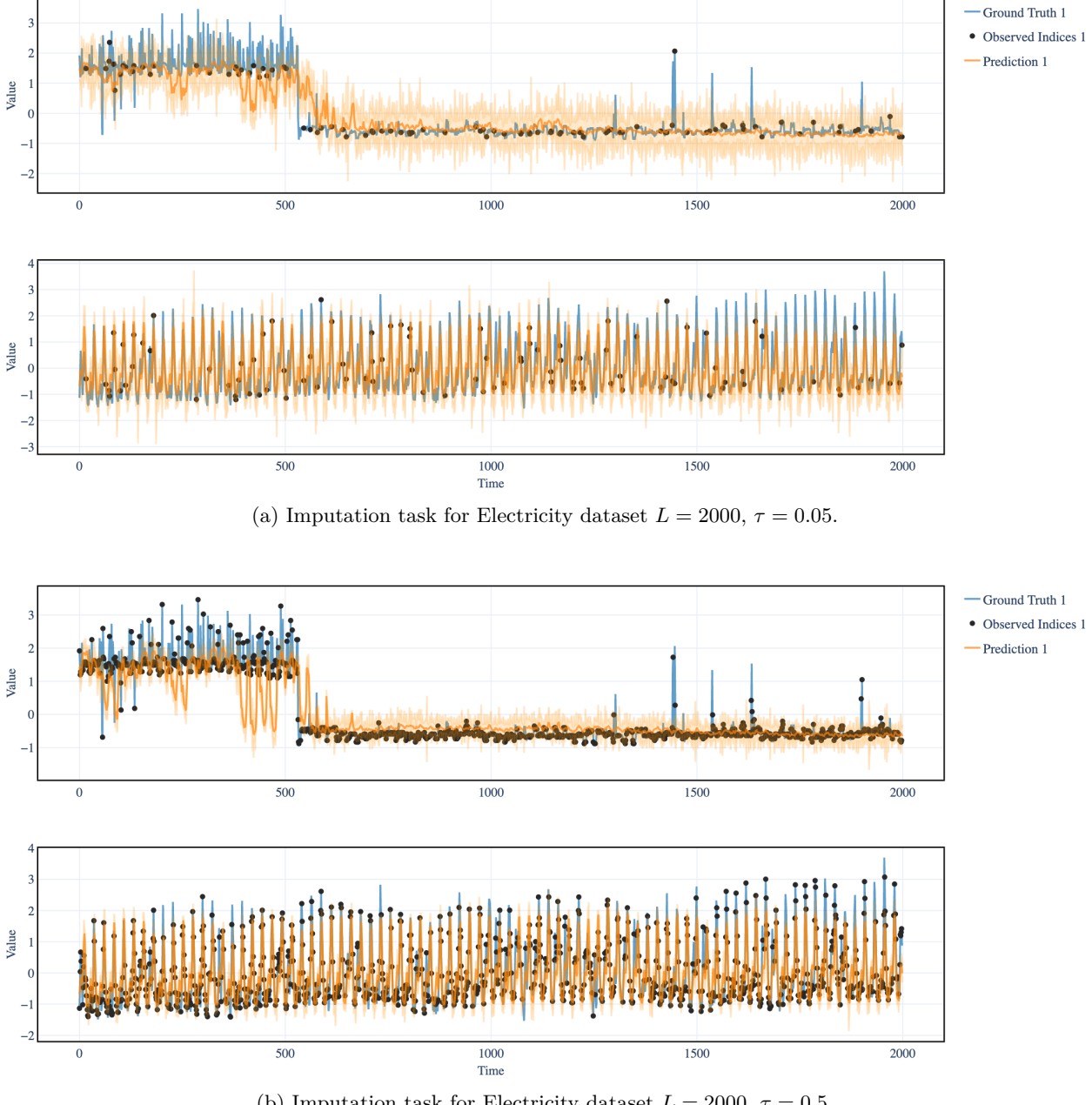

(a) Imputation task for Electricity dataset $L = 2000$, $\tau = 0.05$.

(b) Imputation task for Electricity dataset $L = 2000$, $\tau = 0.5$.

Figure 6: TV-INRs imputation predictions for Electricity dataset ($L = 2000$). Solid lines denote the posterior mean and shaded regions correspond to the 5th–95th percentile intervals.

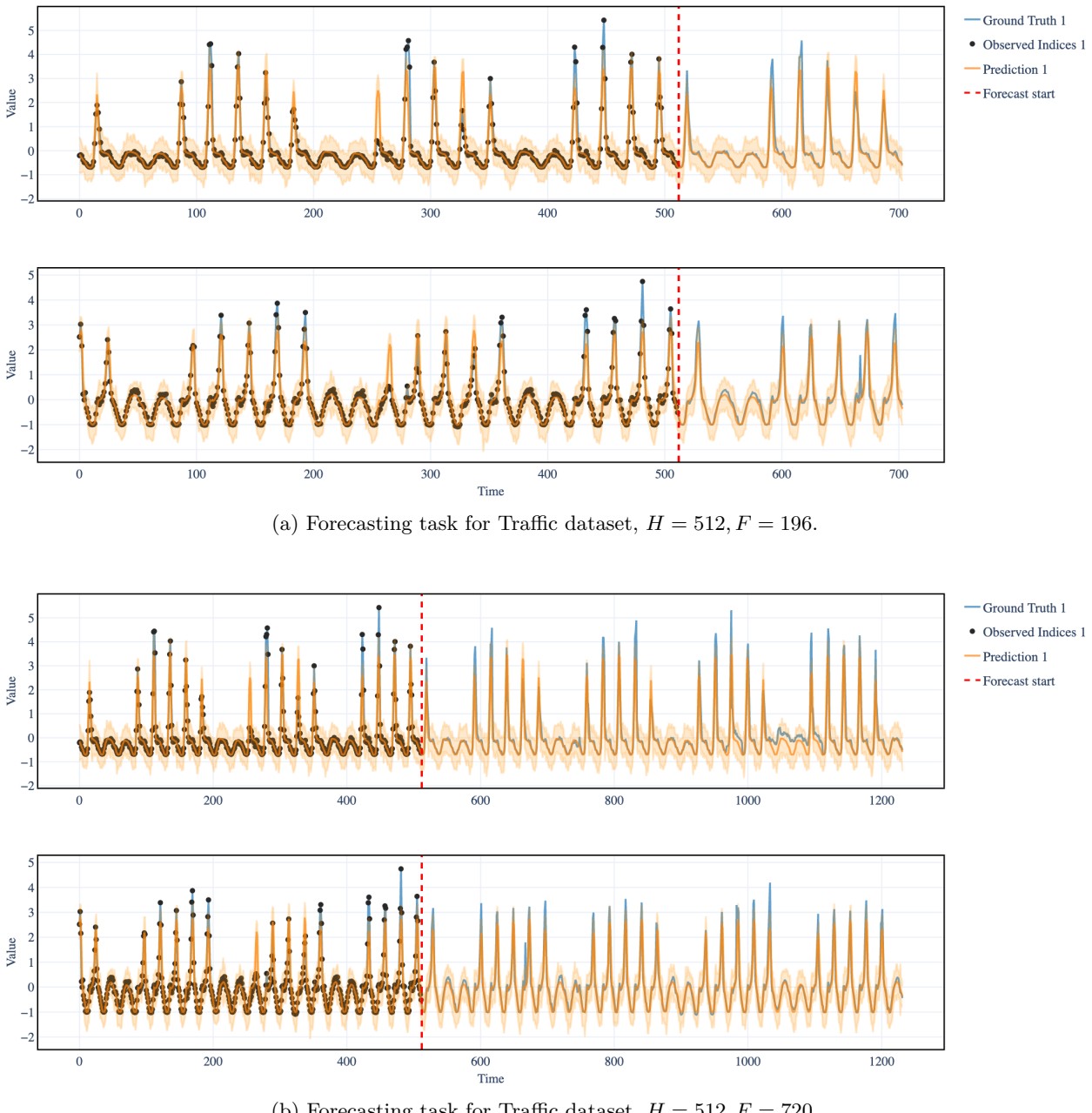

(a) Forecasting task for Traffic dataset, $H = 512, F = 196$.

(b) Forecasting task for Traffic dataset, $H = 512, F = 720$.

Figure 7: TV-INRs forecasting predictions for Traffic dataset. Solid lines denote the posterior mean and shaded regions correspond to the 5th–95th percentile intervals.

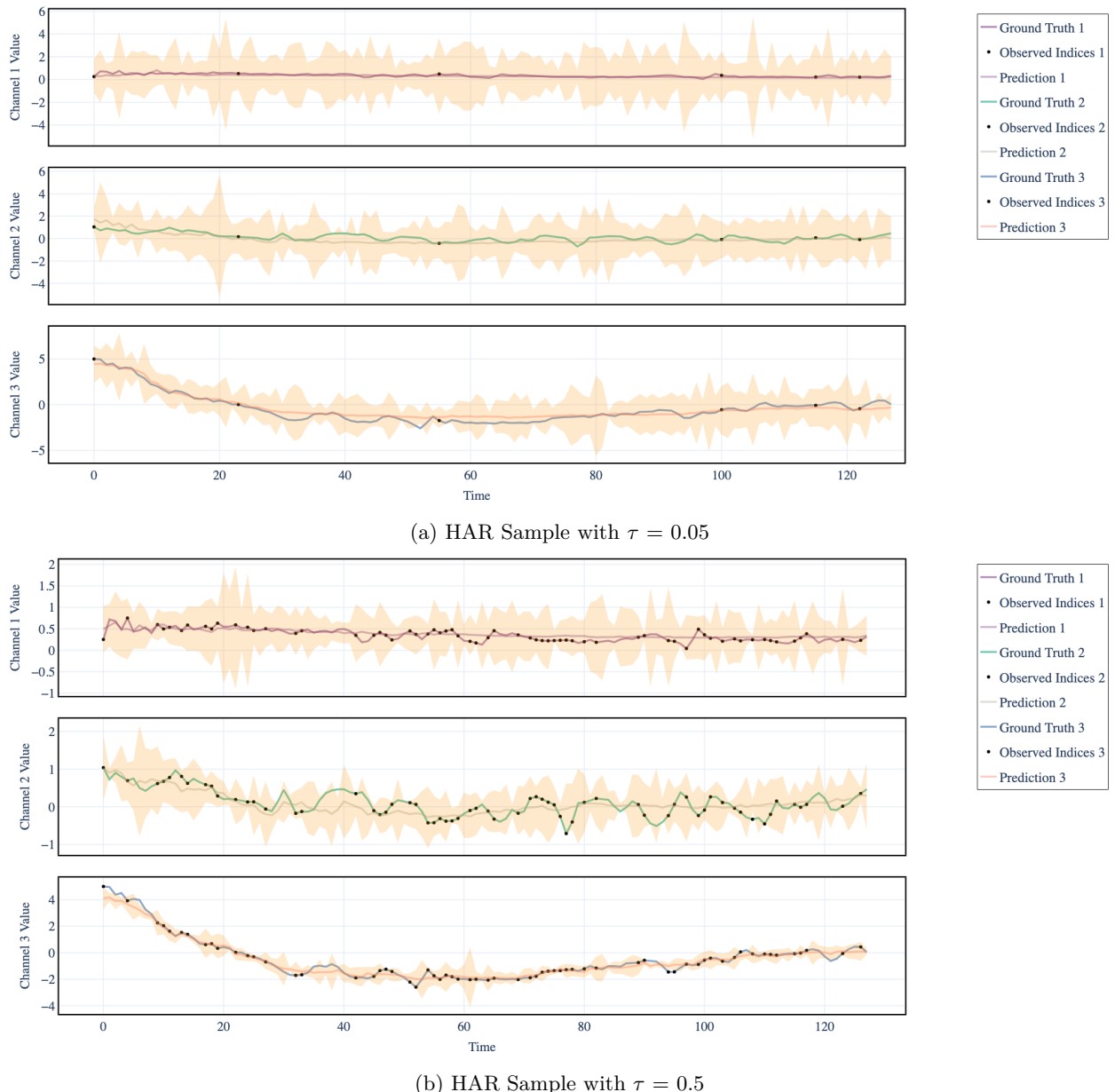

(a) HAR Sample with $\tau = 0.05$

(b) HAR Sample with $\tau = 0.5$

Figure 8: TV-INRs imputations for HAR dataset. Solid lines denote the posterior mean and shaded regions correspond to the 5th–95th percentile intervals.

