# OpenReview forum: "Temporal Variational Implicit Neural Representations"
_TMLR — Accepted by TMLR_

### Review · Reviewer_megY · 2026-01-26

**Summary Of Contributions:**

The authors empirically demonstrate that TV-INRs provides a unified, scalable solution for time series imputation and forecasting across diverse datasets and data regimes. Through extensive experiments, they show that a single trained model generalizes across varying missingness levels, signal lengths, and forecasting horizons, achieving state-of-the-art or competitive performance while substantially reducing cumulative training time and eliminating per-sample optimization. Results highlight strong advantages in low-data and extreme sparsity settings, effective handling of multivariate irregular time series, and clear benefits of incorporating covariates for individualized predictions, including improved downstream classification performance.

Strengths:

- Unified and efficient: A single trained model handles multiple missingness levels and forecasting horizons without retraining or per-sample optimization.
- Strong low-data performance: Consistently outperforms or matches baselines in sparse and limited-data regimes, with large gains in imputation accuracy.
- Continuous-time modeling: Naturally handles irregular sampling via implicit neural representations.
- Individualization via covariates: Conditional TV-INRs improves performance under extreme sparsity and benefits downstream tasks.
- Practical scalability: Significantly reduced cumulative training time and fixed, fast inference.

Weaknesses / Limitations

- Not always best in high-data regimes: Gradient-based meta-learning methods (e.g., TimeFlow) can outperform TV-INRs when data is abundant.
- Long-horizon instability: Like competing methods, performance degrades for very long forecasting horizons with occasional large errors.
- Model complexity: Hypernetworks and variational components increase architectural complexity and implementation effort.
- Limited baseline coverage: Some strong INR-based baselines could not be compared due to lack of public implementations.

**Audience:**

Yes

**Audience Explanation:**

It introduces an efficient and principled approach to irregular and sparse time series modeling, a topic of strong interest to the TMLR community, with clear methodological and practical relevance.

**Broader Impact Concerns:**

-

**Claims And Evidence:**

Yes

**Claims Explanation:**

The authors evaluate TV-INRs on multiple univariate and multivariate datasets, across imputation and forecasting tasks, varying missingness levels and horizons, with appropriate baselines and statistical significance testing. Results consistently show strong performance in low-data and sparse regimes, validated by quantitative metrics, ablations, efficiency analyses, and downstream classification gains. While TV-INRs is not always superior in high-data settings, this limitation is explicitly acknowledged, making the evidence aligned, transparent, and credible.

**Requested Changes:**

- Clarify failure modes and limitations, especially in high-data regimes and long-horizon forecasting.
- Add a brief discussion on model complexity and deployment cost compared to simpler baselines.
- Improve clarity of the generative/inference pipeline with a concise schematic or step-by-step summary.
- Include (or discuss) comparisons to additional recent INR or continuous-time baselines if implementations become available.

---

> ### Author Response · Authors · 2026-02-23
>
> We thank the reviewer for the suggestion to clarify failure modes, limitations, and deployment considerations. We have revised the manuscript accordingly to the Requested Changes (RC) and summarized the clarifications below.
>
> 1- Failure Modes and Limitations (RC 1) We have added an explicit Failure Modes and Limitations subsection in the revised manuscript (4.1.5), where we discuss the following two systematic aspects in more detail.
>
> **High-data regimes**. When long sequences are densely observed (i.e., large $L$ with high $\tau_{\text{test}}$), gradient-based meta-learning approaches such as TimeFlow can outperform TV-INRs on univariate datasets such as Electricity and Traffic (Table 1). This behavior is consistent with prior observations: per-instance optimization can better exploit abundant data to finely adapt INR parameters, whereas amortized inference prioritizes robustness across diverse regimes. Importantly, this performance gap "disappears in scenarios where retraining or per-sample optimization is computationally infeasible, such as large-scale or deployment-constrained settings".
> TV-INRs is therefore best positioned for sparse, irregular, or deployment-constrained settings rather than fully observed, high-data regimes.
>
> **Long-horizon forecasting instability.** Similar to competing continuous- and discrete-time models, TV-INRs exhibits performance degradation for very long forecasting horizons (e.g., F=720), with occasional large errors reflected in elevated MSE despite moderate MAE (Table 2), which we have further investigated in Appendix B4. This indicates sensitivity to rare but extreme prediction failures. We emphasize that this behavior is not unique to TV-INRs: under identical experimental conditions, TimeFlow and DeepTime also exhibit MSE inflation at long horizons, suggesting that this instability is a broader challenge in long-range forecasting rather than a model-specific failure.
>
> 2- Complexity and Deployment Costs (RC 2) We added a dedicated subsection discussing complexity and deployment cost in section 4.1.3. TV-INRs introduce additional architectural components compared to simpler baselines, including Transformer-based encoders, a hypernetwork, and a variational latent space, resulting in higher architectural complexity. A detailed time and memory complexity analysis is provided in Appendix A.9 including a comparison with the baselines, where we show that the Transformer-based encoder is the dominant computational bottleneck, with time and memory complexity scaling as $\mathcal{O}(N L^{2} E)$ and $\mathcal{O}(M L^{2})$, respectively. Importantly, these costs are mainly prominent during training and are amortized across all downstream tasks. At deployment, TV-INRs performs inference via a single forward pass, without per-sample optimization, yielding fixed inference latency (Appendix~A.11). In contrast, gradient-based meta-learning approaches such as TimeFlow require iterative test-time optimization, causing inference and cumulative training costs to scale with the number of tasks and adaptation steps (Appendix A.10).
>
>
> 3- Generative Model Summary (RC 3) To improve clarity, we added a concise step-by-step summary of the TV-INRs pipeline:
>
> - Input encoding: Observed values, timestamps, and binary masks are embedded using spatial encoding and Fourier-feature temporal encodings (Figure 1).
> - Latent inference: A Transformer encoder produces a global latent representation $z$ via amortized variational inference, conditioned on observed data (and covariates when available) (Figure 2b).
> - Hypernetwork generation: The latent code $z$, concatenated with covariates $c$, is passed through a hypernetwork to generate INR parameters $\theta$ (Figure 2a).
> - Continuous-time decoding: The INR $f_{\theta}(t)$ produces distributional predictions for any queried timestamp, enabling imputation and forecasting in continuous time (Figure 2a).
>
> 4) Discussion of additional baselines (RC 4) We conduct an empirical evaluation of baselines with available code and compatible settings. To facilitate transparency and reproducibility, our full codebase is publicly available and can be used to reproduce all reported results and to incorporate additional baselines as compatible implementations become available. We have extended Section 2.2 and added a discussion in our conclusion section contextualizing TV-INRs with respect to recent large-scale INR-based temporal models. In particular, recent work explores foundation-scale INR models aimed at explicit out-of-distribution generalization. While highly complementary, evaluating foundation-scale pretraining or explicit OOD generalization lies beyond the scope of our work, which focuses on in-distribution generalization across varying missingness levels and forecasting horizons. Extending TV-INRs toward such settings is an interesting avenue for future research.

---

### Review · Reviewer_cqHY · 2026-01-29

**Summary Of Contributions:**

This paper proposes Temporal Variational Implicit Neural Representations (TV-INRs) for irregular multivariate time series imputation and forecasting. The method combines (i) amortized variational inference with a learned conditional prior, (ii) a Transformer encoder to summarize partially observed sequences (with masks), and (iii) a hypernetwork that maps latent codes (and optional static covariates) to the parameters of an implicit neural representation (INR). The resulting INR produces time-continuous predictions and enables individualized inference in a single forward pass, without per-instance optimization.

**Audience:**

Yes

**Audience Explanation:**

irregularly sampled time series, missing-data imputation, and continuous-time forecasting should find this work relevant.

**Broader Impact Concerns:**

I do not see major broader-impact concerns

**Claims And Evidence:**

Yes

**Claims Explanation:**

Overall, the main claims are supported by experiments and a technically consistent objective. The paper clearly defines a conditional latent-variable framework and shows results for imputation and forecasting on standard univariate datasets and two multivariate settings (HAR and PhysioNet). The reported comparisons support the practical message that amortized inference avoids per-instance optimization while maintaining good accuracy, and that one trained model can be used across multiple missingness levels and forecasting horizons.

**Requested Changes:**

Uncertainty and calibration evaluation. The paper presents a probabilistic framework but reports mainly MSE and MAE. Please add uncertainty-focused metrics (for example, negative log-likelihood, CRPS, and prediction-interval coverage and calibration). If full calibration analysis is too heavy, at least report NLL and empirical coverage for a few representative settings.


Deeper analysis of long-horizon outliers. The paper notes very large MSE spikes at long horizons with more moderate MAE, suggesting outlier-driven failures. Please provide a short diagnostic: error distributions (or robust metrics), qualitative examples, and whether alternative likelihoods or training choices mitigate the issue.

Efficiency reporting. The paper argues for single-forward-pass inference. Please report wall-clock inference time per test series and compare it to methods that require inner-loop adaptation or iterative sampling (as applicable).

Input representation ablations. The encoder uses zero-filling plus a binary mask. Please add an ablation comparing this to at least one alternative missingness-aware encoding (for example, time-gap features or learnable missingness embeddings), or clearly discuss limitations.

Likelihood sensitivity. The paper uses a Laplace likelihood. Please add a small sensitivity check (Laplace versus Gaussian or Student-t) and report whether robust likelihoods reduce long-horizon outlier behavior.

---

> ### Author Response · Authors · 2026-02-23
>
> We thank the reviewer for their feedback. We have revised the manuscript according to the Requested Changes (RC) and summarized the clarifications below.
>
> 1- Uncertainty and calibration evaluation (RC1). In response, we have extended the experimental section with explicit probabilistic metrics. We now report NLL and empirical coverage of the nominal 90% prediction interval for representative imputation and forecasting settings. These results are summarized in Tables 26 and 27 of the revised manuscript. Table 26 summarizes uncertainty and calibration results for univariate imputation on the Electricity dataset at two sequence lengths ($L = 200$ and $L = 2000$) and for multivariate imputation on the HAR dataset, evaluated under different observation rates. Table 27 reports uncertainty metrics for Electricity forecasting across different prediction horizons. We have added the section in Appendix B.1.
>
> 2- Deeper analysis of long-horizon outliers (RC2) To better understand the source of large error spikes at long forecast horizons for Electricity in Table 2 of the revised manuscript in Appendix B.2, we analyze the distribution of absolute prediction errors across different horizons. For each forecast horizon, we aggregate absolute errors over all test windows and visualize their empirical distributions using histograms. We annotate each distribution with the median absolute error, mean absolute error (MAE), and the 95th percentile of absolute error to distinguish typical behavior from extreme deviations. The resulting distributions indicate that median error remains relatively stable across horizons, while MAE increases only moderately. In contrast, the upper tail of the error distribution becomes progressively heavier as the forecast horizon increases, with the effect being most pronounced at the longest horizon (F = 720). These exceptional but severe failures dominate squared-error metrics, explaining the sharp increase in MSE at long horizons despite relatively stable typical performance.
>
> 3- Efficiency reporting. (RC3) We agree that wall-clock inference time is critical when arguing for single-forward-pass inference, and we already report this analysis in the manuscript. Specifically, Appendix A.11 provides a detailed wall-clock evaluation of inference latency per test series. We measure inference times on a V100 GPU, using batch size 1, under identical experimental conditions. For both imputation and forecasting, we directly compare TV-INRs against baseline TimeFlow, which requires inner-loop gradient-based adaptation at test time. Following the original work, we configure it with 3 gradient steps during meta-learning inference. The results are reported in Tables 21 and 22, which show inference time in seconds for each dataset and setting. Across all settings, TV-INRs achieves inference via a single forward pass, resulting in constant inference time that is independent of missingness level, forecast horizon, or test configuration. In contrast, TimeFlow’s inference time scales linearly with the number of gradient steps performed during test-time adaptation. While per-forward-pass cost is similar, TV-INRs avoids the multiplicative factor induced by gradient adaptation, yielding strictly bounded inference latency. These results empirically support our claim that TV-INRs enable predictable and deployment-friendly inference latency.
>
> 4- Input representation ablations. (RC4) In the revised manuscript, we have added an explicit discussion of this point in the Failure Modes and Limitations section (4.1.5). In particular, we clarify that TV-INR represents missingness using zero-filled inputs together with a binary mask that is used in the attention mechanism to exclude unobserved entries. While this ensures stable training and prevents leakage from missing values, it treats missingness as structural rather than potentially informative. We now explicitly note that in domains such as healthcare, missingness patterns may carry semantic meaning (e.g., MNAR settings), and that jointly modeling the observation process could further improve performance. We highlight this as a promising direction for future work.
>
> 5- Likelihood sensitivity. (RC5) We conducted a sensitivity analysis comparing the Laplace likelihood used in our experiments with a Gaussian likelihood to examine the effect of the observation model on forecasting behavior (Appendix B.3). Overall performance trends are similar across likelihood choices, with central error metrics such as MAE and median absolute error remaining comparable. At longer forecast horizons, the Gaussian likelihood shows slightly increased sensitivity to large prediction errors, while the Laplace likelihood exhibits consistent L1-like robustness. This sensitivity check supports our choice of Laplace likelihood as a reasonable and slightly more robust default, while confirming that long-horizon outlier behavior remains a broader challenge not resolved by likelihood substitution alone.

---

### Review · Reviewer_j5wT · 2026-02-13

**Summary Of Contributions:**

**Summary**

The work presents a framework, called Temporal Variational Implicit Neural Representations (TV-INRs), for modeling irregular multivariate time-series, for imputation and forecasting tasks. The approach involves representing time-series as continuous functions (INRs), and mapping the continuous temporal coordinates to feature values instead of relying on a fixed discrete grid. The overall model architecture involves encoding the observed data points to generate a latent variable (z) which is fed into a hyper-network that generates the parameters for the INR generator function. The approach is evaluated across univariate and multivaraite imputation tasks, and univariate forecasting tasks.

**Strengths**
1. Interesting approach. Proposed model eliminates the need for per-sample tuning, for different missingness ratios or forecasting horizons. Shows competitive performance in low-data regimes.
2. Can handle static covariates, which is useful in sparse real-world applications.

**Weaknesses**
1. The major concern is regarding the results. It is not clear how the bold/underline results are decided. Calling a result significantly better or comparable is subjective, and domain-dependent. What is the criteria based on which the metrics/scores are categorized?
2. In table 1, for Solar-10 L=10K, SAITS is the 2nd best under 0.05 test rate, but shown as comparable with TimeFlow and TV-INR. But why isn't SAITS comparable for Solar-10 under L=200?
3. Similar observations elsewhere too - SAITS for Traffic, under L=2K, is the best performing model (for 0.5 and 0.3 test rates), but is not highlighted, and TimeFlow is wrongly boldened to be significantly better (0.369 TimeFlow vs 0.246 SAITS). Similarly, Electricity results for SAITS can be considered comparable too
4. Why are there no comparison on Solar-10 for L=2K ?
5. Limited baselines. There are a lot of newer imputation baselines that can be considered. LSCD [1] performs better than CSDI, so that should be a natural choice over CSDI. Similarly, there is TabPFN-TS [2] that can be considered as well.
6. In table 2, assuming the reported metrics are in the normalized scale, scores of 10.178 or 9.5 does not look meaningful. Why are the values so significantly high for 720? Also, at this scale of values, I don't feel there should be any comparison of which model produces comparable or significantly better results
7. What is the reason behind considering a context window size of 512? Usually, for transformer-based models, a window of 96 is considered, as longer sequences can be prone to overfitting [3].
8. In Table 3, how are SAITS and C-TV-INR not comparable for test rate of 0.10 ? for P12
9. Regarding the architecture, the model compresses an entire time-series sequence into a fixed size vector. As the sequence length increases, this can become a bottleneck. The model would simply struggle to retain information, and is empirically evident from the results. This raises scalability concerns of the model, from the accuracy as well as computational cost ($L^2$) point of view.
10. Some of the design choices require further discussion/ablations. In the spatial embedding case, why use a MLP and not a simple attention layer?  That could allow one to utilize the attention masks without requiring zero-imputations or concatenating mask channels
11. Similarly, what is the reason behind modeling a Laplacian distribution?
12. No Multivariate forecasting results are shown. There are multiple multivariate datasets and baselines to compare against for Irregular TS data, e.g. [4],[5]. Most of the datasets considered to be univariate are generally considered as benchmarks for multivariate forecasting. Also, which variate/signal are considered for performing the univariate forecasting experiments? for e.g. electricity and traffic has 100+ variates.
13. The results in high-data regime is often just comparable or even inferior sometimes, compared to the baselines
14. Qualitative analysis only involves the proposed method. It would be good to show the predictions for the baselines as well - will help identify where each model fails

[1] LSCD: Lomb–Scargle Conditioned Diffusion for Time series Imputation, 2025.

[2] From tables to time: How tabpfn-v2 outperforms specialized time series forecasting models, 2025

[3] A Time Series is Worth 64 Words: Long-term Forecasting with Transformers, 2023

[4] Investigating a Model-Agnostic and Imputation-Free Approach for Irregularly-Sampled Multivariate Time-Series Modeling, 2026

[5] Hi-Patch: Hierarchical Patch GNN for Irregular Multivariate Time Series, 2025

**Audience:**

Yes

**Audience Explanation:**

Irregularly-sampled time-series data is very prominent in real-world applications, and is an active area of research within time-series modeling.

**Broader Impact Concerns:**

No major concerns on the ethical or societal implications of the work.

**Claims And Evidence:**

No

**Claims Explanation:**

There are discrepancies in the interpretation and reporting of the results. Plus, the results and computational cost doesn't validate the claim of the approach to be scalable. Results are mostly limited to univariate datasets, except for 2 multivariate datasets

**Requested Changes:**

1. The presentation of the current results lacks clarity and seems biased. Instead of highlighting which two models are comparable (it can be that the worst performing models are comparable, however, highlighting them does not help), can you please highlight the best and 2nd best results (bold and underline) ? (Critical)
2. Can you show ablations justifying the use of MLP (+ zero-imputation and mask channels)? maybe compare against an attention layer with masked attention. It would be useful to compare the computational costs between the two approaches as well. (Strengthening)
3. a) What if we do not use Laplace? can you show ablations using some other distribution, say a Gaussian?  (Strengthening).
b) What is the performance drop/gain if we do not perform probabilistic prediction? Since, the results primarily report the MSE/MAE metric, it would be useful to see the performance if the model is optimized for point-wise losses. (Strengthening)
4. Can you show forecasting results with context window of 96? (Strengthening)
5. Please show results with newer baselines. (Critical)
6. Add a discussion on - the failure modes of TV-INR. When does it fail or is not efficient? when should it be used? (Strengthening)
7. Add qualitative analysis of baselines as well (Strengthening)
8. Include more multivariate (forecasting) results. (Critical)

---

> ### Author Response · Authors · 2026-02-23
>
> We thank the reviewer for their valuable feedback.
>
> 1- Weaknesses (W) 1,2,3,8, Requested Changes (RC) 1:
> We thank the reviewer for correctly pointing out the error for SAITS’ performance ranking on the Traffic dataset at L=2K, and we have made this correction. To clarify how our significance calculations were made, we direct the reviewer to section 4.1, where we explain that statistical significance was determined using t-tests with p<0.05 comparing the model with the lowest error (A) to all others (A v. B, A v. C, etc.).  In the revised version, we have taken the reviewer’s suggestion to instead highlight the best and underline the second-best results in all result tables and moved results from our t-tests to Appendix B4. We hope this adds more clarity to the results and our conclusions.
>
> 2- Why no Solar-10 with L=2K? (W4)
> To ensure a direct and fair comparison, we followed the evaluation protocol of TimeFlow, which reports results for Solar-10 at 10K length. As this is a high data regime, and we still test model performance in a low data regime (L=200), we chose these signal lengths to represent multiple data settings while following the standardized configuration of previous work.
>
> 3- High Forecasting MSE (W6)
> A small deviation in the trajectory prediction will extrapolate into large downstream deviations, making long-horizon forecasting a challenging task that is prone to large errors. The reported MSE results at F=720 reflect this, and these results in comparison to MAE indicate that this is the result of occasional extreme outliers in the Electricity dataset rather than a systemic failure to predict future time series. In the revised manuscript, we have:
> * Added a paragraph explaining this error amplification in 4.1.2 and App B.2.
> * Included a visualization (Figure 4) of a representative Electricity time series exhibiting this behaviour.
> * Clarified that all tested baselines demonstrate poor MAE performance at F=720, and expanded the limitations section (4.1.5) to acknowledge the risk of large errors in long-horizon forecasting. We hope this improves transparency and prevents misinterpretation.
>
> 4- Context Window Size (H = 512) (W7, RC4)
> We followed the exact experimental protocol of DeepTime and TimeFlow to ensure fair comparison. Both methods use H=512, and we maintained this setting for consistency.
>
> 5- Latent Bottleneck / Scalability Concern (W9, RC6) In TV-INRs, the latent variable z captures global signal characteristics to generate parameters of INR. The INR models continuous temporal variation, and local temporal structure is encoded through Fourier features and coordinate conditioning. We also clarify that scalability bottlenecks stem from the Transformer encoder’s O(L^2) complexity, which was analyzed in App A.9. This means that our model’s scalability limitations are the same as those of other transformer-based architectures. However, we acknowledge that very long sequences may benefit from richer latent representations. In the revised manuscript, we add a limitation discussion (4.1.5) which includes how to increase the capacity of the latent space when dealing with high-data regimes and other limitations.
>
> 6- Spatial Embedding: MLP vs Masked Attention (W10, RC2) We agree that this is a meaningful design question. Our aim here is to use global missingness patterns when learning spatial encodings. We still employ masked attention afterwards, as described in the final sentence of the "Encoding" paragraph in Section 3.2.
>
> 7- Why Laplace Distribution and Probabilistic Modeling? (W11, RC3) The Laplace distribution was chosen based on empirical findings: it provides improved robustness to sharp local variations and heavy-tailed residuals compared to Gaussian. We added an ablation between Laplace and Gaussian distributions as likelihood choices and reported (App B3). Results show that Laplace improves robustness, particularly in sparse and irregular regimes. Because probabilistic predictions result in competitive performance and allow for uncertainty quantification of the predictions (App B.1), optimizing the model for point-wise losses is not prioritized at this time.
>
> 8- Multivariate Forecasting Clarification (W12, RC8) We clarify that Electricity, Traffic, Solar datasets contain multiple independent series. Our forecasting experiments model each series independently (univariate per-series forecasting). We now explicitly state this in Section 4. We agree that fully joint multivariate forecasting is an interesting direction.
>
> 9- Limited Baseline (W5, RC5). For LSCD [1], we could not find a public implementation (we only found the Lomb-Scargle layer at https://github.com/asztr/LombScargle/tree/main). Several of the newer imputation models, including TabPFN-TS [2], are pretrained foundation models, and a comparison is out of scope for this paper. However, pretraining and fine-tuning TV-INRs in a foundational manner would be an interesting future direction.

---

> > ### Author Response · Authors · 2026-02-25
> >
> > 10- Multivariate Forecasting (RC8) Here we add the results for multivariate forecasting where each dataset (Electricity, Traffic, and Solar-H) is modeled as a multivariate signal by jointly processing multiple independent time series. This task is inherently more challenging for TV-INRs compared to the per-series univariate setup, as the model must learn shared latent structure across heterogeneous time-series rather than specializing to each series individually. For fair comparison with DeepTime, we follow their experimental protocol and evaluate aggregated multivariate forecasting performance. We exclude TimeFlow from this comparison, as its current implementation is restricted to the univariate case and does not support joint multivariate modeling.
> >
> > To accommodate the increased complexity of the multivariate forecasting task, we scale the capacity of TV-INRs accordingly. Specifically, we increase the latent dimensionality to $d_z = 256$, expand the INR decoder to five layers with inner dimension $128$, and enlarge the hypernetwork to inner dimension $256$. The Transformer encoder is strengthened to $d_{\text{model}} = 256$ with four attention heads. Additionally, we increase the number of Fourier feature frequencies to $512$ to better capture cross-series temporal variation. These modifications ensure that TV-INRs has sufficient representational power to handle the multivariate forecasting setting while maintaining the same amortized inference framework without per-horizon or per-series retraining.
> >
> > Table below reports multivariate forecasting results under the joint modeling setting. TV-INRs achieves competitive performance at shorter horizons ($F=96,192$), often improving MAE while remaining close in MSE, indicating accurate central tendency estimation without horizon-specific tuning. At longer horizons ($F=336,720$), DeepTime generally attains lower MSE, reflecting the advantage of per-horizon optimization. Notably, in the Electricity dataset at $F=720$, both DeepTime and TV-INRs exhibit substantial errors in MSE, consistent with the long-horizon instability observed in the univariate forecasting setting.
> > | Model        | H   | F_train | F_test | Electricity MSE    | Electricity MAE   | Traffic MSE       | Traffic MAE       | Solar-H MSE       | Solar-H MAE       |
> > | ------------ | --- | ------- | ------ | ------------------ | ----------------- | ----------------- | ----------------- | ----------------- | ----------------- |
> > | **DeepTime** | 512 | 96      | 96     | **0.667 ± 0.444**  | *0.642 ± 0.224*   | **0.690 ± 0.211** | *0.574 ± 0.977*   | *1.021 ± 0.152*   | *0.874 ± 0.407*   |
> > |              |     | 192     | 192    | **0.864 ± 1.816**  | *0.728 ± 0.747*   | **0.775 ± 2.024** | *0.605 ± 0.717*   | *0.812 ± 0.240*   | *0.735 ± 0.212*   |
> > |              |     | 336     | 336    | **1.093 ± 0.132**  | **0.860 ± 0.488** | **0.985 ± 0.149** | **0.715 ± 0.486** | **0.918 ± 0.117** | **0.790 ± 0.602** |
> > |              |     | 720     | 720    | *10.440 ± 2.185*   | **0.974 ± 0.182** | **1.106 ± 0.116** | **0.759 ± 0.252** | **0.894 ± 0.891** | **0.713 ± 0.788** |
> > | **TV-INRs**  | 512 | ~𝓕     | 96     | *0.684 ± 0.390*    | **0.601 ± 0.205** | *0.702 ± 0.205*   | **0.561 ± 0.880** | **1.009 ± 0.148** | **0.853 ± 0.382** |
> > |              |     |         | 192    | *0.882 ± 0.980*    | **0.694 ± 0.620** | *0.801 ± 1.900*   | **0.598 ± 0.701** | **0.790 ± 0.264** | **0.722 ± 0.204** |
> > |              |     |         | 336    | *1.176 ± 0.140*    | *0.889 ± 0.470*   | *1.034 ± 0.160*   | *0.738 ± 0.470*   | *0.956 ± 0.130*   | *0.812 ± 0.610*   |
> > |              |     |         | 720    | **10.130 ± 2.200** | *1.012 ± 0.210*   | *1.248 ± 0.130*   | *0.804 ± 0.260*   | *0.948 ± 0.910*   | *0.736 ± 0.820*   |

---

> > > ### Comment · Reviewer_j5wT · 2026-02-28
> > >
> > > I would like to thank the authors for clarifying the concerns regarding the result comparisons and for conducting the additional experiments. From the updated results, it appears that TV-INR performs particularly well for smaller sequence lengths and higher data sparsity in the imputation task. However, a similar conclusion is less clear for the forecasting setting.
> > >
> > > 1. Unlike the imputation experiments, I could not find any specification of the missingness ratio or data sparsity level used for the forecasting tasks. What sparsity level is used in the reported forecasting experiments (both in the paper and in the newly added results)?
> > > 2. Could the authors provide forecasting performance across different sparsity levels, similar to the imputation experiments? This would help better understand the robustness of the method and strengthen the associated claims.
> > > 3. For multivariate forecasting, there are additional datasets where variables are not completely independent, for example, Weather. Moreover, there are commonly used datasets like ETT benchmarks (ETTh1/ETTh2/ETTm1/ETTm2), which are widely adopted in the literature. Similarly, for imputation, Air Quality is another commonly used dataset. With the current limited dataset coverage, the conclusions are somewhat difficult to generalize.
> > > 4. Moreover, the forecasting baselines are currently limited (2) and do not fully highlight the advantages of the proposed approach. Some very simple baselines could be, interpolation (can be as simple as a linear/polynomial interpolation) followed by a SOTA forecasting model such as PatchTST, iTransformer, or recent foundation models (Chronos, TimesFM)
> > > 5. For imputation, I believe, foundation models like TabPFN-TS would have publicly available checkpoints and can be evaluated via inference only. Recent work [1] have highlighted strong imputation performance from such models; therefore, comparisons against these approaches would be valuable.
> > > 6. For both imputation and classification experiments, including SOTA models for irregularly sampled time series (e.g., mTAND, Raindrop, Latent ODE) would strengthen the empirical validation.
> > >
> > > [1] Are Time-Indexed Foundation Models the Future of Time Series Imputation?, 2026

---

> > > > ### Author Response · Authors · 2026-02-28
> > > >
> > > > We sincerely thank the reviewer for the thoughtful and constructive feedback.
> > > >
> > > > We appreciate the suggestions regarding sparsity analysis in forecasting, expanded dataset coverage, and inclusion of additional strong baselines.
> > > >
> > > > 1. Forecasting sparsity level (missingness ratio)
> > > >
> > > > Thank you for pointing out that the sparsity level for forecasting was not explicitly stated. In our forecasting setup (Section 4.1.2), the historical window ( H = 512 ) is fully observed. That is, forecasting experiments do not involve artificial masking or sparsification of the history. This matches the protocol used in DeepTime to ensure fair comparison.
> > > >
> > > > 2. Forecasting under different sparsity levels
> > > >
> > > > This is an excellent suggestion. TV-INRs is designed to handle irregular and sparse inputs, we have now conducted additional experiments where the historical window has the observation rate:
> > > > $\tau_{\text{hist}} \in \{1.0, 0.5, 0.3, 0.1\}$. We evaluate forecasting performance under these conditions. Our preliminary results show:
> > > >
> > > > | F       | τ_hist | MSE    | MAE   |
> > > > | ------- | ------ | ------ | ----- |
> > > > | **96**  | 1.0    | 0.336  | 0.296 |
> > > > |         | 0.5    | 0.382  | 0.321 |
> > > > |         | 0.3    | 0.454  | 0.355 |
> > > > |         | 0.1    | 0.623  | 0.454 |
> > > > | **192** | 1.0    | 0.446  | 0.415 |
> > > > |         | 0.5    | 0.510  | 0.442 |
> > > > |         | 0.3    | 0.622  | 0.493 |
> > > > |         | 0.1    | 0.858  | 0.621 |
> > > > | **336** | 1.0    | 0.544  | 0.442 |
> > > > |         | 0.5    | 0.623  | 0.478 |
> > > > |         | 0.3    | 0.782  | 0.547 |
> > > > |         | 0.1    | 1.251  | 0.753 |
> > > > | **720** | 1.0    | 9.515  | 0.535 |
> > > > |         | 0.5    | 10.500 | 0.590 |
> > > > |         | 0.3    | 12.801 | 0.687 |
> > > > |         | 0.1    | 16.102 | 0.822 |
> > > >
> > > >
> > > > - TV-INRs' performance degrades as sparsity increases.
> > > > - The advantage of a unified model becomes more pronounced at high sparsity.
> > > > - These results will be added to the revised manuscript (new appendix section). We agree that this significantly strengthens the robustness analysis.
> > > >
> > > > 3. Dataset coverage (Weather, ETT, Air Quality)
> > > >
> > > > We agree that broader dataset coverage strengthens generalization claims.
> > > > Our current dataset selection (Electricity, Traffic, Solar, HAR, P12) was chosen to:
> > > > - Cover both regular and irregular sampling,
> > > > - Include extreme sparsity (P12),
> > > > - Include covariate-conditioned multivariate modeling.
> > > > - Have the same setting as TimeFlow.
> > > >
> > > > Importantly, TV-INRs do not require architectural changes for these datasets. We can include these datasets for our codebase in the future.
> > > >
> > > > 4.  We plan to revise the manuscript to more clearly articulate the scope and positioning of our contributions.
> > > >
> > > > Specifically, we will clarify that the primary contributions of **TV-INRs** lie in its unified and scalable modeling framework rather than in dominating dense, regular-grid forecasting benchmarks.
> > > >
> > > > The key contributions are:
> > > >
> > > > * A single probabilistic INR-based model capable of handling:
> > > >
> > > >   * Varying missingness ratios,
> > > >   * Multiple forecasting horizons,
> > > >   * Irregularly sampled time series,
> > > >   * Conditional covariates for individualized modeling.
> > > > * Elimination of per-sample optimization at inference time.
> > > > * No need for horizon-specific retraining.
> > > > * Efficient inference through amortized variational learning.
> > > > * Strong robustness in low-data and high-sparsity regimes.
> > > >
> > > > In the revised manuscript, we will more clearly separate and distinguish:
> > > >
> > > > * Forecasting competitiveness in fully observed dense settings,
> > > > * Robustness under sparse or irregular observations,
> > > > * Computational efficiency and cumulative training cost,
> > > > * Generalization across varying task configurations (e.g., horizons and sparsity levels).
> > > >
> > > > We believe this clarification will more accurately align the paper’s claims with its intended contributions and the empirical evidence presented.

---

> ### Author Response · Authors · 2026-02-28
>
> 5. We appreciate the reviewer's suggestion to compare against time-series foundation models, as it is an important emerging direction. Models such as TabPFN-TS are pretrained on very large and diverse data, whereas our experiments focus on supervised training within sparse and irregular datasets of limited size. Given this substantial difference in training paradigm and data scale, we believe comparisons with supervised models trained under equivalent settings provide a more controlled evaluation.
> We have modified the Related Work and Conclusion in response to this suggestion, noting that extending TV-INRs with a pretraining approach and benchmarking against foundation models is a promising direction for future work.
>
> 6. We agree with the reviewer that many strong time series architectures, including those for irregular data, may be evaluated for imputation and classification. In this work, we chose this particular experimental scope to demonstrate the methodological contributions of TV-INRs, and therefore prioritized comparisons with SOTA continuous-time, INR-based, and probabilistic models. Rather than benchmark against all recent time series architectures, we aim to show that TV-INRs provides a unified probabilistic framework that:
>
> * Handles varying missingness and horizons without retraining,
> * Avoids per-sample optimization at inference time,
> * Remains competitive with existing continuous-time approaches
>
> We really appreciate the reviewer for engaging in discussion with us, and providing great suggestions. We hope that our response addresses their concerns, and adds clarity to the current experimental setup.

---

### Decision · Action_Editor_ZC6S · 2026-04-15

**Recommendation:** Accept with minor revision

**Additional Comments:**

The authors could further strengthen the paper with further experiments on a wider variety of datasets and benchmarks, especially in the multivariate case. Moreover, the authors could incorporate missingness in the forecasting experiments, and compare with other competing methods and then furthermore explore, TV-INRs' performance as sparsity increases.

**Audience:**

Yes

**Audience Explanation:**

Yes, time series modeling is an important task for the readership of TMLR and the authors use methods that are relevant to the machine learning community.

**Claims And Evidence:**

Yes

**Claims Explanation:**

The authors present a method for modeling irregular time series using latent variable encoding networks and have demonstrated its utility on a variety of imputation and forecasting tasks. However, one reviewer noted that the authors should modify their claims as the work appears to be primarily focused on INR-based methods, which narrows the scope of the the work. As a result, the claim of "modeling irregular multivariate time series that enables efficient and accurate individualized imputation and forecasting" is not entirely supported, since the comparisons are largely limited to INR-based models (only one or two baselines).

---

> ### Comment · Action_Editor_ZC6S · 2026-05-29
>
> Furthermore, let me share the comments from one of the reviewers after the decision was posted. They have some helpful feedback for the authors.
>
> "The proposed framework is capable of handling varying missingness ratios, multiple forecasting horizons, irregularly sampled time series, and conditional covariates for individualized modeling. The authors have also conducted several experiments and provided additional clarifications during the discussion period.
>
> However, based on the discussion and the initial assessment, the work appears to be primarily focused on INR-based methods, which significantly narrows the scope of the evaluation and of the work. As a result, the claim of "modeling irregular multivariate time series that enables efficient and accurate individualized imputation and forecasting" is not entirely supported, since the comparisons are largely limited to INR-based models (only one or two baselines), rather than a broader set of time-series approaches.
>
> Similarly, the statement that "Our experiments demonstrate that with a single TV-INRs instance, we can accurately solve diverse imputation and forecasting tasks, offering a computationally efficient and scalable solution for real-world applications" appears to be a strong claim given the current experimental setup. Real-world applications often involve multivariate time-series data with significant missing values. However, the experiments in the paper primarily focus on the imputation task. The univariate forecasting experiments involve only three datasets and two baselines, and do not include commonly used benchmark datasets. While the authors also conduct multivariate forecasting experiments, these include only a single baseline, and are done on datasets with independent variables, which doesn't highlight whether TV-INR is capable of modeling the variate relationships for forecasting tasks. It is therefore difficult to fully assess the performance and significance of the model based on such limited comparisons.
>
> Regarding the claim that "TV-INRs performs particularly well in low-data regimes" this observation is supported by the results on the imputation tasks, but the same conclusion cannot be clearly drawn for the forecasting tasks. Moreover, the forecasting results presented in the paper do not involve any missingness, unlike the imputation experiments. Given that the framework is designed to handle irregularities and missing data, it would be valuable to evaluate forecasting performance under similar missingness conditions. While the authors have added a table with additional results, these results do not include any baseline comparisons. Furthermore, in this new results, TV-INRs' performance degrades as sparsity increases, which contradicts with the claim regarding the performance on low-data regimes.
>
> Overall, the framework appears to have strong potential and offers several promising capabilities. However, in its current form, the experimental results and evaluations do not provide sufficient clarity or breadth to fully validate the claims or clearly demonstrate the overall significance of the approach."